# Prosit-XL: enhanced cross-linked peptide identification by fragment intensity prediction to study protein interactions and structures

Mostafa Kalhor [1], Cemil Can Saylan [1], Mario Picciani [1], Lutz Fischer [2], Falk Boudewijn Schimweg [2], Joel Lapin [1], Juri Rappsilber [2,3,4] & Mathias Wilhelm [1,5] ✉

It has been shown that integrating peptide property predictions such as fragment intensity into the scoring process of peptide spectrum match can greatly increase the number of confidently identified peptides compared to using traditional scoring methods. Here, we introduce Prosit-XL, a robust and accurate fragment intensity predictor covering the cleavable (DSSO/DSBU) and non-cleavable cross-linkers (DSS/BS3), achieving high accuracy on various holdout sets with consistent performance on external datasets without fine-tuning. Due to the complex nature of false positives in XL-MS, an approach to data-driven rescoring was developed that benefits from Prosit-XL's predictions while limiting the overestimation of the false discovery rate (FDR). After validating this approach using two ground truth datasets consisting of synthetic peptides and proteins, we applied Prosit-XL on a proteome-scale dataset, demonstrating an up to ~3.4-fold improvement in PPI discovery compared to classic approaches. Finally, Prosit-XL was used to increase the coverage and depth of a spatially resolved interactome map of intact human cytomegalovirus virions, leading to the discovery of previously unobserved interactions between human and cytomegalovirus proteins.

Crosslinking mass spectrometry (XL-MS) has emerged as a critical technology for analyzing protein complexes and protein-protein interactions (PPIs) by providing distance restraints between protein residues through the identification of cross-linked peptides (XL-peptides)[1]. However, several main computational challenges arise from this process. The tandem mass spectrum (MS2) of XL-peptides is more complex than linear peptides due to the presence of two peptides, making sequence assignments challenging. Unequal fragmentation of the two peptides could introduce bias in the total cross-linked peptide

spectrum match (CSM) score[2]. Another challenge is the large search space caused by considering all possible peptide pairs (n-square problem), which can increase the chance of false positives before false discovery rate (FDR) estimation. To maintain a fixed FDR, stricter score thresholds are required, leading to higher false negatives after FDR filtering and ultimately reducing the number of true positives. In addition, the FDR estimation using the target-decoy approach for XL-peptides is complicated by the presence of two peptides and remains a challenge in XL-MS research[3–5]. Numerous crosslinking database

---

[1]Computational Mass Spectrometry, TUM School of Life Sciences, Technical University of Munich, Freising, Germany. [2]Bioanalytics, Institute of Biotechnology, Technical University Berlin, Berlin, Germany. [3]Wellcome Centre for Cell Biology, University of Edinburgh, Edinburgh, UK. [4]Si-M/"Der Simulierte Mensch", a Science Framework of Technische Universität Berlin and Charité - Universitätsmedizin Berlin, Berlin, Germany. [5]Munich Data Science Institute, Technical University of Munich, Garching, Germany. ✉e-mail: mathias.wilhelm@tum.de

search engines (XL-DBSEs) have been developed over the past decade to address these challenges, including xiSEARCH[6], pLink2[7], XlinkX[8], Kojak[9], Scout[10], etc. However, identifying XL-peptides remains challenging, especially for inter-protein crosslinks (between-links) where the search space size is significantly larger than for intra-protein crosslinks (self-links)[11]. One approach to improving the scoring process of DBSE for linear peptide identification is to use post-processing rescoring tools such as Percolator[12] and PeptideProphet[13,14]. It aims to integrate multiple DBSE features (e.g., DBSE scores, peptide length) into a single score for FDR calculation[15]. The XL-MS field is no exception, with XL-DBSEs like XlinkX and Kojak applying Percolator, while pLink 2 uses a built-in SVM classifier as a rescoring tool. However, evaluating these XL-DBSEs on multiple ground truth datasets has revealed that applied rescoring tools often suffer from overfitting, leading to suboptimal accuracy in FDR calculation[10,16,17].

In addition to DBSE features, integrating accurate peptide property predictors, such as Prosit[18,19], a fragment intensity predictor, and DeepLC[20], a retention time (RT) predictor, into the rescoring process has been shown to substantially increase the number of confidently identified linear peptides compared to relying only on DBSE features. This improvement is particularly evident in challenging fields such as immunopeptidomics[19], single-cell proteomics[21], and metaproteomics[18]. In the XL-MS field, tools like xiRT[22], a retention time predictor for XL-peptides, and pDeepXL[23], a predictor of fragment ion intensity for XL-peptides, have been developed to enhance XL-peptide identifications. However, these tools have limitations, including (1) often requiring fine-tuning for rescoring implementation, (2) yielding modest improvement in CSM and PPI level identification rates after rescoring, and (3) needing a more user-friendly design for rescoring tasks. Additionally, further investigation using ground truth datasets is necessary to fully validate their effectiveness.

In this study, we introduce Prosit-XL, an expanded version of Prosit[19], developed through transfer learning for fragment intensity prediction of XL-peptides. We propose an approach by considering each XL-peptide as two separate peptides, allowing for data augmentation by using each CSM twice during training. Additionally, Prosit-XL inherited its collision energy (CE) awareness from Prosit, circumventing the need for transfer learning on unseen data. We have integrated Prosit-XL into our user-friendly, data-driven rescoring pipeline Oktoberfest[24]. It has been shown that the lower score of the two peptides in a XL-peptide can serve as a strong parameter for distinguishing correct from incorrect matches[2,9]. Here, we adopt a similar approach but introduce a novel aspect by running Percolator on the peptide spectrum match (PSM) level, here referring to the individual peptide of an XL-peptide, rather than CSM level, referring to an entire XL-peptide. We aggregate the intensity-based scores into a single score by using the minimum Percolator-optimized PSM-level score of the two PSMs associated with a CSM as a proxy for its quality, a strategy that more effectively separates correct from incorrect matches, resulting in a substantial boost of identified cross-linked peptides. Prosit-XL and the rescoring pipeline are validated on full and partial ground truth datasets containing synthetic peptides and proteins, respectively. Next, we benchmark Prosit-XL's performance on a combined large-scale dataset containing *E. coli* and *M. pneumoniae*, comparing its performance to xiSEARCH/xiFDR. Ultimately, we apply Prosit-XL and the rescoring pipeline on real-world data from intact human cytomegalovirus to demonstrate its capability in increasing the depth and coverage of XL-peptides required for comprehensive protein structure and PPI discovery and analysis.

## Results

### Accurate fragment intensity prediction by Prosit-XL

Due to the absence of synthetic data on the scale required for our applications, we had to rely on public data. Multiple public XL-MS datasets from PRIDE[25] were downloaded, focusing on cleavable

(disuccinimidyl sulfoxide, DSSO[26–28]; disuccinimidyl dibutyric urea, DSBU[29]) and non-cleavable (disuccinimidyl suberate, DSS[26,28,30–32]; bis-sulfosuccinimidyl suberate, BS3[30]) cross-linkers. Datasets containing MS2 spectra of cleavable (CMS2) and non-cleavable (NMS2) XL-peptides were analyzed by pLink 2, while XlinkX was used for MS3 spectra of cleavable XL-peptides (CMS3). This resulted in 125,727 CMS2, 70,320 NMS2, and 37,938 CMS3 identified high-quality spectra containing ~31,000, ~17,000, and ~9000 unique XL-peptides (peptide pairs), respectively (Fig. 1a). The CMS2 and NMS2 spectra were acquired by higher-energy collisional dissociation (HCD) fragmentation, while CMS3 spectra were obtained using collision-induced dissociation (CID).

As highlighted earlier[33], a main factor that can substantially affect fragment ion intensities using HCD fragmentation is the normalized CE (NCE), which can vary even on the same mass spectrometer despite using the same ostensible CE, due to drifts of the effective NCE applied. To enable an NCE-dependent prediction of fragment intensities, acquired data needs to be calibrated. This is particularly difficult for XL-MS datasets due to the lack of replicate peptide spectra across datasets, which could have been used to detect drifts in NCE. Therefore, we proposed to detect shifts in NCE using contaminant linear peptides within the datasets. These identified peptides are then utilized to estimate the NCE at which the predictions match best to the acquired spectra (NCE calibration). This estimation is achieved by comparing the spectra of the top highest scoring PSMs to predictions made by the HCD Prosit 2020 model at different NCEs (Methods). The NCE at which the highest normalized spectral angle (SA) was observed indicates the optimal NCE for prediction (Fig. 1b). The optimal NCEs for NMS2 and CMS2 spectra fall within the ranges of 28 to 40 and 17 to 37, respectively (Fig. 1c, d). Note that NCE calibration is not necessary for CSM3 spectra which were acquired using CID fragmentation.

The collected XL-MS data is substantially smaller than the dataset used for training Prosit, which was trained using the ProteomeTools synthetic peptide library[34]. This is in part due to the complex nature of analyzing XL-peptides, e.g., challenging fragmentation, identification, and lower abundance in samples. The collected CSMs are equivalent to only ~2% of the PSMs used for training the HCD Prosit 2020 model[19] (Supplementary Fig. 1a). This is not sufficient to train a model from scratch because that would require a substantial reduction in model size. We propose using transfer learning, the same approach used in developing pDeepXL, where a pre-trained model on a similar task is adapted and further trained for a new, related task. Here, we use Prosit[19] as a starting point to extend it and apply transfer learning for XL-peptides.

In order to make an informed decision on the required adjustments of the Prosit model, two crucial questions need to be answered: Are the fragment ion intensities of peptide A influenced by a cross-linked peptide B, and, if so, are its intensities dependent on the sequence of peptide B? To delve into this, we first compared Prosit's predictions to fragment ion intensities of XL-peptides. The results (Supplementary Fig. 1b) show that Prosit's predictions agree somewhat with intensities extracted from CMS3 spectra (e.g., median SA of ~0.51 for Alkene-CMS3), but performs very poorly on CMS2 (e.g., median SA of ~0.35 for DSSO-CMS2) and NMS2 (e.g., median SA of ~0.40 for DSS/BS3-NMS2). This suggests that the presence of a peptide B influences the fragmentation characteristics more than the presence of a cross-linker. However, Prosit's performance is still better than random, estimated by Prosit's performance on decoy XL-peptides (median SA of ~0.01, Supplementary Fig. 1c). Second, we assessed the similarity of MS2 fragment intensity patterns of XL-peptides, where the XL-peptides share the same peptide A but differ in peptide B (Fig. 1e), for many different peptides A. Taken together, XL-peptides do indeed influence each other's fragmentation characteristics and thus implies that peptide B must be considered as a separate input to a model in order to reach good prediction performance. This is not the case for

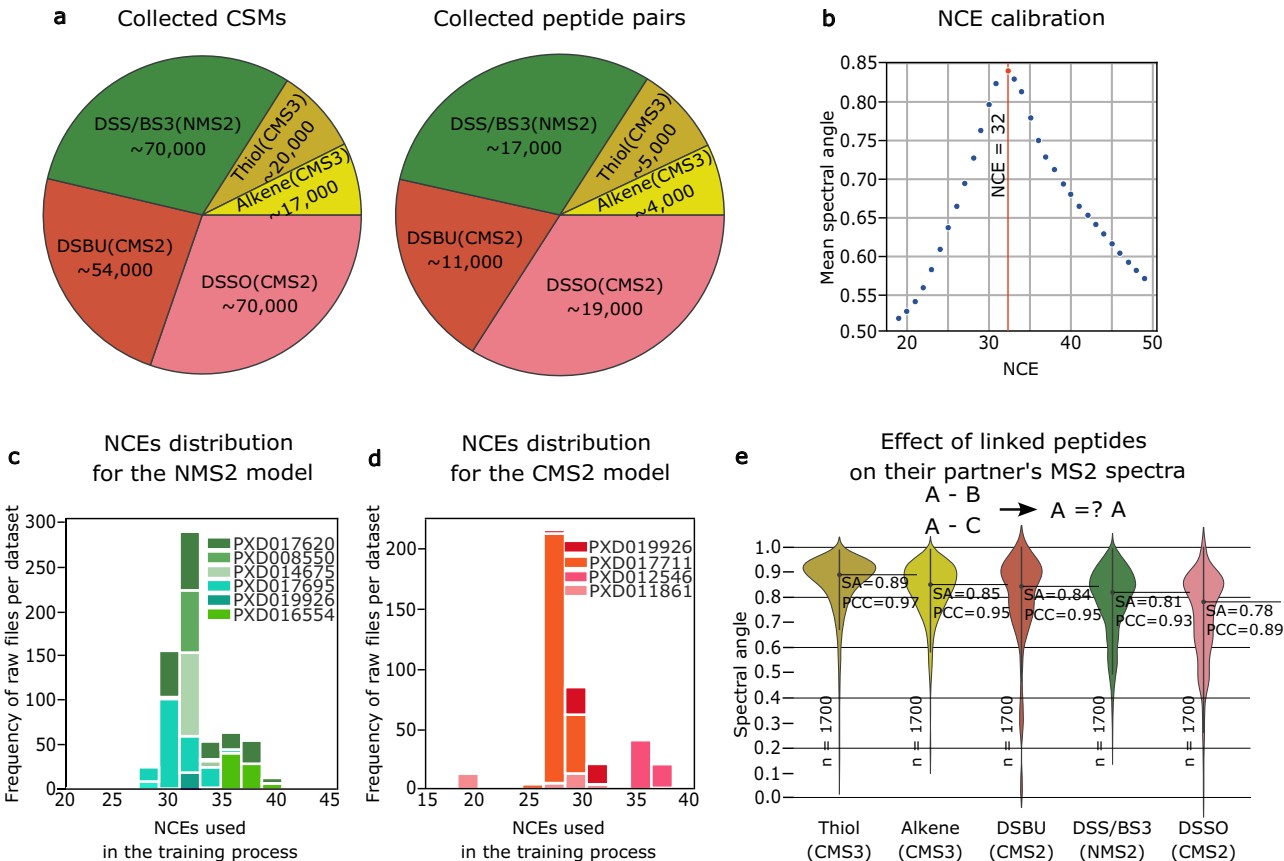

**Fig. 1 | Data collection and collision energy calibration for refining Prosit to Prosit-XL. a** Pie chart showing the collected training data on cross-linked spectrum match (CSM) and unique XL-peptide (peptide pair) level, covering the cleavable cross-linkers Thiol and Alkene acquired using MS3 spectra (CMS3) and DSSO and DSBU acquired using MS2 spectra (CMS2), as well as data for non-cleavable cross-linkers DSS/BS3 acquired using MS2 spectra (NMS2). **b** Normalized collision energy (NCE) calibration curve for an example MS-file, showing the mean spectral angle (SA) when comparing annotated experimentally acquired spectra of the top 1000 highest scoring target peptide-spectrum matches (PSMs) to spectra predicted with Prosit at varying NCEs. The NCE with the highest average SA, indicated by the vertical red line, is used as the NCE for training. **c, d** Bar plots showing the number of raw files after NCEs calibration across the training data. **e** Violin plot comparing the annotated experimental MS2-MS3 spectra of XL-peptides with the same peptide A but different peptide B for five different cross-linkers: CMS3-Thiol, CMS3-Alkene, CMS2-DSBU, NMS2-DSS/BS3, and CMS2-DSSO. The analysis is not focused on a specific peptide A; instead, peptide A refers to the first peptide in each cross-linked pair. The number of sampled spectra ($n = 1700$) is indicated at the bottom. The black solid line and corresponding numbers indicate the median spectral angle (SA) and Pearson correlation (PCC) for each distribution. Mean spectral angles ± standard error of the mean (SEM) for each group are as follows: Thiol(CMS3), 0.857 ± 0.003; Alkene(CMS3), 0.827 ± 0.003; DSBU(CMS2), 0.779 ± 0.005; DSS/BS3(NMS2), 0.785 ± 0.003; and DSSO(CMS2), 0.724 ± 0.004.

CMS3, since only one of the two peptides is fragmented to acquire a CMS3.

In order to accurately reflect the requirements of the various XL-peptides, various adjustments to the architecture of Prosit are required (Fig. 2a). One additional encoder (Encoder 2) was added to handle the input of peptide B for both Prosit-XL-CMS2 and Prosit-XL-NMS2 models. Moreover, an extra decoder (Decoder 2) was added for Prosit-XL-CMS2 since DSSO and DSBU have two unstable sites, resulting in two distinct types of XL fragments: b-short and y-short, with a shorter segment of the crosslinker, and b-long and y-long, with a longer segment of the crosslinker. In order to maximize the benefits of transfer learning, Prosit-XL is trained to predict only the fragment ion intensities of peptide A, whereas the encoder for peptide B only "modulates" the intensities of peptide A. This further allowed us to utilize each spectrum in the training and test set twice, resulting in a twofold increase in the effective training dataset size. To obtain predictions for peptide B, peptides A and B are swapped. A schematic representation of the best-performing Prosit-XL-CMS2 and Prosit-XL-NMS2 architecture with more details is shown in Supplementary Fig. 2a. Since CMS3 does not require any modifications to the base architecture, the architecture of Prosit-XL-CMS3 is the same as that of CID Prosit 2020 model.

After finalizing the architecture of these models, we initialized the Prosit-XL-CMS2 model with the weights from the HCD Prosit 2020 and trained it on the collected and calibrated CMS2 data. Subsequently, the Prosit-XL-NMS2 was initialized with the weights from the Prosit-XL-CMS2 and trained on collected and calibrated NMS2 data (Supplementary Fig. 2b). The Prosit-XL-CMS3 model was initialized using the CID Prosit 2020 model weights and trained on the corresponding CMS3 data. After the training process, the median SA on the holdout set improved by 0.31 − 0.48, depending on the type of spectra (Fig. 2b). It should be emphasized that SAs are measured separately for peptide A and B. Interestingly, the Prosit-XL model's performance on peptide B(s) is slightly better than peptide A(s), likely due to differences in peptide lengths. On average, peptide B is shorter than peptide A which is generally less challenging for the Prosit-XL to predict (Supplementary Fig. 2c). For benchmarking Prosit-XL's generalization, Prosit-XL-CMS2 and Prosit-XL-NMS2 models were applied, without any additional transfer learning, to two distinct external datasets using synthetic peptides[16,17] that were cross-linked by DSSO and DSS. Initially, xiSEARCH and pLink were used to identify CSMs. To ensure optimal prediction performance, NCE calibration was performed to find the optimal NCE for each MS file, which was then used as input for the model (Methods). The Prosit-XL-CMS2 achieved a median SA of 0.82

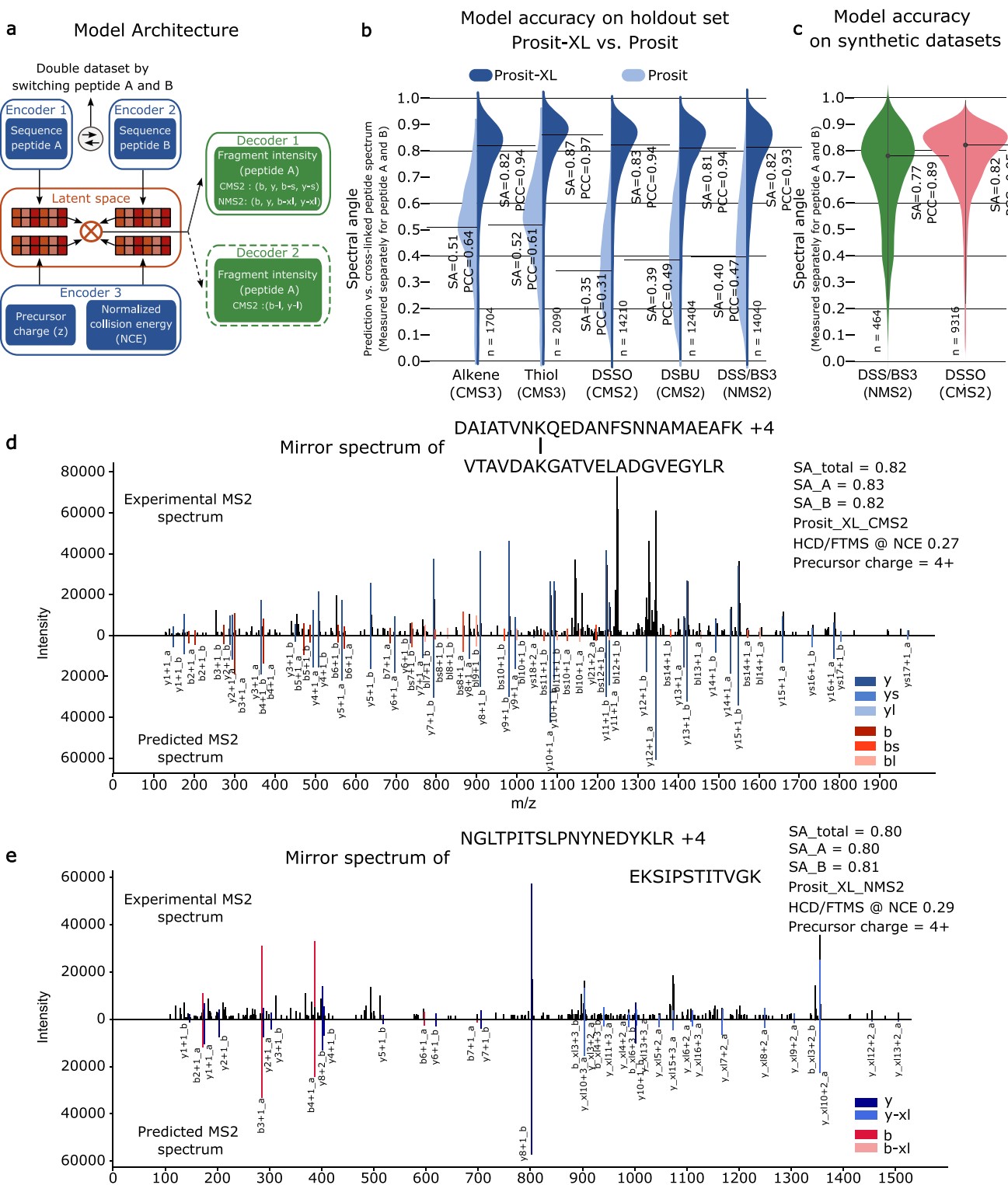

and a Pearson correlation (PCC) of 0.95, showing its remarkably consistent performance compared to the holdout set (Fig. 2c). The Prosit-XL-NMS2 is slightly below that of the holdout set (median SA: 0.77, median PCC: 0.89), which may indicate additional experimental factors not considered in the proposed architecture of Prosit-XL. These results suggest that no additional transfer learning is needed for Prosit-XL to achieve state-of-the-art performance.

Further, pDeepXL was applied to these two datasets to compare the performance of Prosit-XL and pDeepXL. On CSMs supported by both models, Prosit-XL demonstrated higher accuracy, achieving an SA of 0.82 and a PCC of 0.95, compared to pDeepXL, which achieved an SA of 0.74 and a PCC of 0.85, on the dataset that used DSSO as a crosslinker. However, both models showed almost identical performance on the synthetic dataset that used DSS as a crosslinker, where Prosit-XL achieved an SA of 0.75 and a PCC of 0.90, while pDeepXL achieved an SA of 0.76 and a PCC of 0.87 (Supplementary Fig. 2d, Methods). To visually demonstrate the Prosit-XL's performance, two mirror plots of a cleavable and a non-cleavable XL-peptide are

**Fig. 2 | Accurate fragment ion intensity prediction of XL-peptides by Prosit-XL.**
**a** Schematic illustration of the general architecture of Prosit-XL-CMS2 and Prosit-XL-NMS2 for fragment ion intensity prediction of XL-peptides. The input data (XL-peptide precursor charge state, normalized collision energy (NCE), peptide sequence A, and peptide sequence B) are encoded into a latent representation (latent space). These representations are then element-wise multiplied and subsequently decoded to fragment ion intensities. Prosit-XL-CMS2 contains one extra decoder compared to Prosit-XL-NMS2 covering y-long and b-long fragments. The Prosit-XL-CMS3 has the same architecture as HCD Prosit 2020, missing the Encoder 2 and Decoder 2. **b** Violin plot comparing the prediction accuracy of Prosit-XL models (dark blue) for CMS3, CMS2, and NMS2 compared to the prediction accuracy of the previously published HCD Prosit 2020 and CID Prosit 2020 model (light blue) on the holdout set across 5 different cross-linker types: CMS3-Alkene, CMS3-Thiol, CMS2-DSSO, CMS2-DSBU, and NMS2-DSS/BS3. The number of underlying spectra (n) is indicated at the bottom. The black solid line and corresponding numbers indicate the median spectral angle (SA) and Pearson correlation

coefficient (PCC). The prediction performance was assessed separately for peptides A and B (PSM level). **c** Violin plot demonstrating the prediction accuracy of Prosit-XL-CMS2 and Prosit-XL-NMS2 on external unseen datasets using DSSO and DSS/BS3 as cross-linkers. The number of underlying spectra (n) is indicated at the bottom. The black solid line and corresponding numbers indicate the median spectral angle and Pearson correlation. Data are presented as mean ± SEM: DSSO (mean = 0.776, SEM = 0.002), DSS/BS3 (mean = 0.726, SEM = 0.008). **d, e** Mirror spectrum of two XL-peptides comparing the experimentally acquired spectrum (top spectrum) to its respective prediction by Prosit-XL for the peptide DAIATVNKQEDANFSNNAMAEAFK (peptide A) cross-linked by DSSO with VTAV-DAKGATVELADGVEGYLR (peptide B) predicted by Prosit-XL-CMS2 (**d**) and the peptide NGLTPITSLPNYNEDYKLR (peptide A) cross-linked by DSS with EKSIP-STITVGK (peptide B) predicted by Prosit-XL-NMS2 (**e**). Matching peaks are visualized in dark red, red, and light red for b, b-s, and b-l ions, respectively, and in dark blue, blue, and light blue for y, y-s, and y-l, respectively.

displayed in Fig. 2d, e showing strong agreement between experimental (top spectra) and predicted fragment ion intensities of the annotated b- and y-ions (bottom spectra).

## Evaluating Prosit-XL with ground truth benchmark datasets

In line with observations on linear peptides[18,19,35,36], we hypothesized that integrating fragment intensity predictions into CSM scoring would improve the differentiation between true and false positive CSMs in target-target (TT) identifications, leading to substantial improvement in the confident identification of CSMs, at a given FDR threshold, compared to utilizing XL-DBSE scores alone. To test this, we extended our open-source data-driven rescoring pipeline, Oktoberfest[24], for CSM rescoring, which consumes predictions from the Prosit-XL models that are served by Koina[37], an open-source online prediction service. Briefly, the rescoring process (Fig. 3a) starts with reading MS files and unfiltered search results of supported XL-DBSEs (Method). Using unfiltered search results allows Oktoberfest to reassess all provided matches, including previously rejected (non-confidently identified) but potentially true positive identifications. After NCE calibration, Oktoberfest calculates a plethora of intensity-based features, separately calculated for peptide A and B (Supplementary Table 1), assessing the similarity of spectra predicted at the optimal NCE and the corresponding experimental spectra.

However, for FDR estimation in XL-MS, a single score is required that reflects the quality of a CSM in order to effectively separate correct from incorrect matches. While tools such as Percolator are optimized to combine multiple features, as generated by, e.g., Oktoberfest, into a single score, the complex nature of false positives (e.g., target-decoy matches) and often an insufficient number of matches for robust machine learning in XL-MS prevent its direct application.

We propose an approach in which we use Percolator solely to generate an optimized score for each peptide precursor in an XL-peptide separately by running it on PSM level, rather than on CSM level, which is possible because Prosit-XL generates predictions for each peptide separately. When splitting up CSMs into two separate PSMs (one for peptide A and B each), the clear notion of a target and decoy match remains. Further, the overall PSM-level score distribution of matches follows the expected behavior as known for linear peptides (Supplementary Fig. 3a). The result of this is a score that is optimized to separate correct from incorrect PSMs. Because a CSM is incorrect when at least one of the two PSMs is incorrect, we pick the minimum Percolator-optimized PSM-level scores of the two PSMs associated with a CSM as a proxy for the quality of that CSM (Supplementary Note 1 and Supplementary Fig. 4). Finally, the CSMs and their corresponding scores are submitted to xiFDR for FDR estimation (Methods).

We evaluated our rescoring pipeline on two recently published, distinct full and partial ground truth XL-MS datasets to verify if the FDR estimates are well calibrated. One such dataset contains synthetic peptides, which are grouped and cross-linked by DSSO. Each link between synthetic peptides from different groups or unknown peptides is considered as false positive, allowing precise determination of the experimentally validated proportion of discoveries that are accepted but deemed to be incorrect (actual FDR)[17]. We first analyzed the data using xiSEARCH followed by xiFDR, resulting in the identification of 1395 CSMs and 789 peptide pairs at an estimated FDR of 1% on CSM- and peptide pair-level, while the actual FDR was 1.18% and 1.65% at the CSM and peptide pair levels, respectively (Fig. 3b). By rescoring pipeline, the number of identified CSMs and peptide pairs modestly improved by 14% (to 1,591) and 12% (to 884) on CSM and peptide pair levels, respectively. However, we also observed an increase in actual FDR to 2.53% at CSM and 3.67% at the peptide pair level. The slightly worse accuracy in FDR estimation may be the result of the small dataset size, which contains only 100 synthetic ground truth peptides. Overall, the results are in line with (and largely below) the reported FDR estimates of other software, including MeroX[38], MS Annika[39], XlinkX, pLink 2, MaxLynx[40], and xiSEARCH/xiFDR, whose 1% FDR estimates result in an actual FDR of 5.7%, 2.7%, 4.4%, 4.0%, 2.2%, and 3.2% at the unique residue pair (UXL) level (Methods), respectively. Further, we compared our rescoring pipeline to xiSEARCH by applying an actual FDR of 1%. The results improved after rescoring, with the number of identified CSMs increasing from 1,175 to 1,216 and peptide pairs increasing from 641 to 651, respectively (Supplementary Fig. 3b, Methods).

To further investigate the FDR estimate, we applied rescoring on a larger and more recent dataset[10]. Briefly, this dataset contains hundreds of recombinant proteins that were separately mixed and cross-linked by DSSO. Besides comparing CSM and peptide pair FDR estimates, this dataset also enabled us to verify the PPI-level FDR estimates of the rescoring pipeline. Since we did not have access to Scout's FDR calculator as a standalone tool, we applied xiFDR on both Scout's unfiltered results (Scout+xiFDR) and rescoring results (Scout+Prosit-XL+xiFDR) in order to ensure a fair comparison. Filtered at 1% FDR, both pipelines produced less than 1% false positives (FP), estimated by the known incorrect interactions, of 0.53%, 0.57%, and 0.67% for rescoring and 0.63%, 0.66%, and 0.96% for Scout+xiFDR on CSM-, peptide pair-, and PPI-level (Fig. 3c), respectively, at an applied FDR of 1% on CSM-, peptide pair-, and PPI-level (Methods). Encouragingly, rescoring was able to increase the number of CSMs, PeptideParis, and PPIs (between-links) by 34.9%, 33.4%, and 42.7%, respectively, compared to Scout+xiFDR. Next, we compared the rescoring results with Scout using its native FDR estimation (without xiFDR). Although Scout identified 24.9% more PPIs compared to the rescoring, the actual FDR for PPIs uniquely identified by Scout was 3.7%, whereas the PPIs identified only by rescoring were 0% (Supplementary Fig. 3c). The FDR for PPIs identified by both methods was 0.76%.

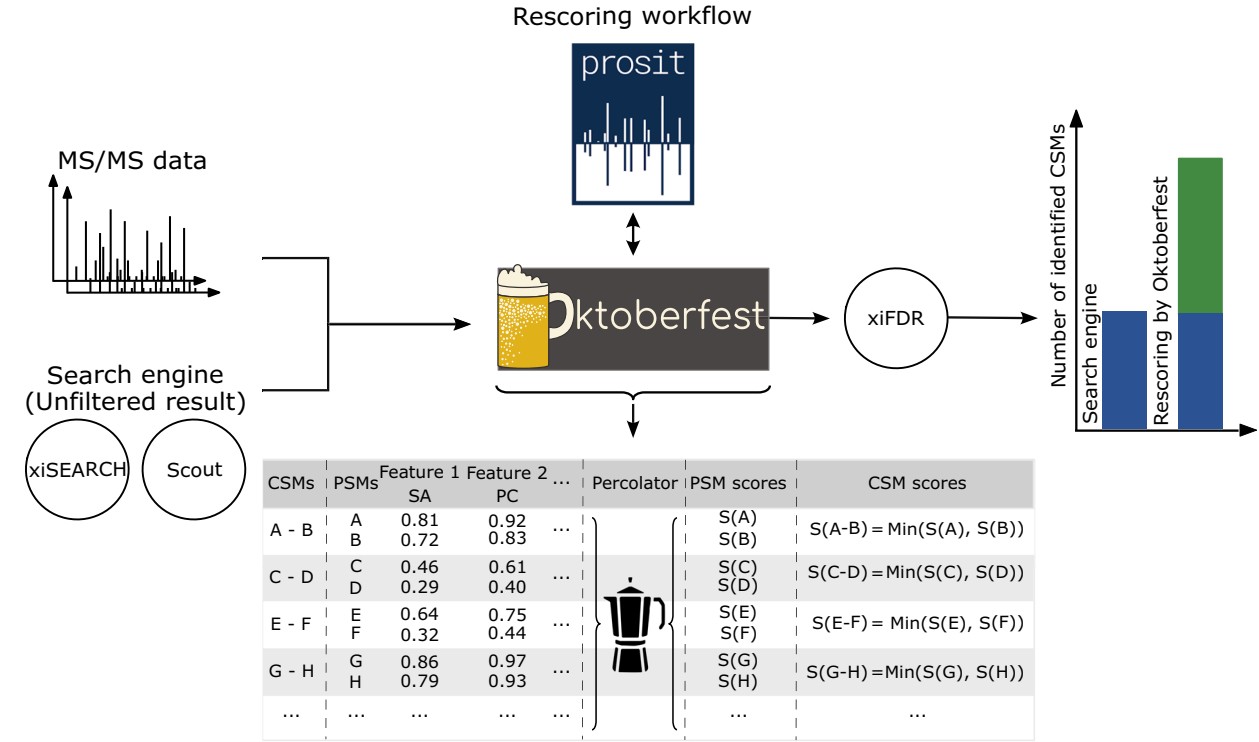

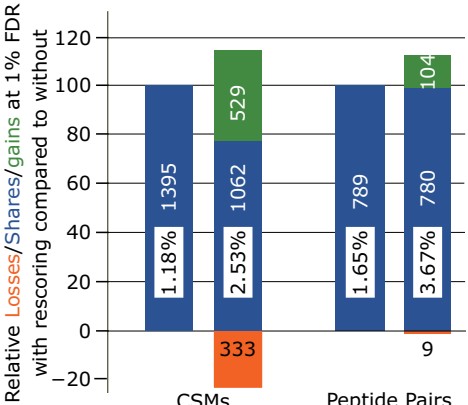

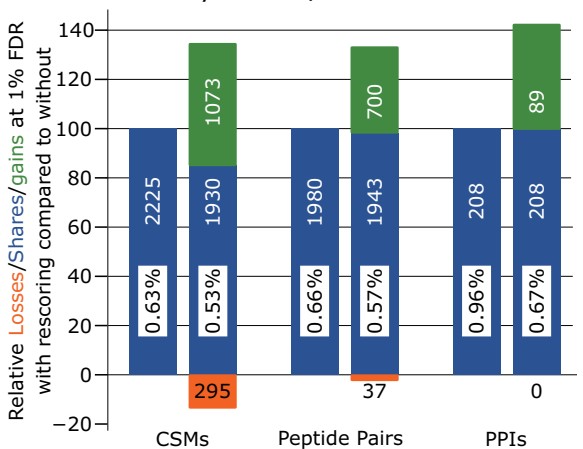

**Fig. 3 | Overview of the rescoring pipeline and its results on ground truth datasets. a** Schematic illustration of the data-driven rescoring pipeline based on Prosit-XL as implemented in Oktoberfest. First, unfiltered results from supported XL-DBSEs (xiSEARCH or Scout) and mass spectrometry (MS) files (e.g., RAW) are required as input for rescoring. Oktoberfest performs spectrum annotation, normalized collision energy (NCE) calibration, and retrieves fragment ion intensity predictions from Prosit-XL to generate an extensive set of intensity-based features for each CSM provided by the XL-DBSE search results. Percolator is run at PSM level (rather than at CSM level). The final CSM score is obtained by taking the minimum percolator discriminant score of each PSM in a CSM and is submitted to xiFDR for FDR estimation on CSM-, peptide pair-, and PPI-level. **b** Vennbars show the number of identified CSMs and peptide pairs lost (orange), shared (blue), and gained

(green), at an FDR of 1% on CSM- and peptide pair-levels when comparing results from xiSEARCH+Prosit-XL+xiFDR to xiSEARCH+xiFDR on a synthetic peptide dataset. Percentages inside the bars represent the actual FDRs, estimated by the ground truth synthetic peptide dataset. The analysis is based on both self- and between-link comparisons. Source data are provided in Supplementary Data 2. **c** Vennbars show the number of identified CSMs, peptide pairs, and PPIs lost (orange), shared (blue), and gained (green) at an FDR of 1% on CSM-, peptide pair-, and PPI-level when comparing results from Scout+Prosit-XL+xiFDR to Scout+xiFDR on a synthetic protein dataset. Percentages inside the bars represent the actual FDRs. The analysis is based on between-links only. Source data are provided in Supplementary Data 3.

## Evaluating Prosit-XL with large-scale datasets and extensive search space

Next, we proceeded to benchmark the rescoring pipeline against an even larger dataset that resembles the quality and complexity of real data more closely. To be able to retain some level of control over the FDR estimation, we re-analyzed two distinct XL-

experiments, investigating PPIs in *E. coli*[5] and *M. pneumoniae*[41], in a combined xiSEARCH run (Fig. 4a). Any identified *E. coli-M. pneumoniae* PPI, *E. coli-E. coli* PPI supported by spectra from the *M. pneumoniae* dataset and *M. pneumoniae - M. pneumoniae* PPI supported by spectra from the *E. coli* dataset must be considered false positives (mismatch) and thus provide a lower bound estimate of

**a**

Workflow for FDR evaluation in
large-scale *E. coli* and *M. pneumoniae* datasets

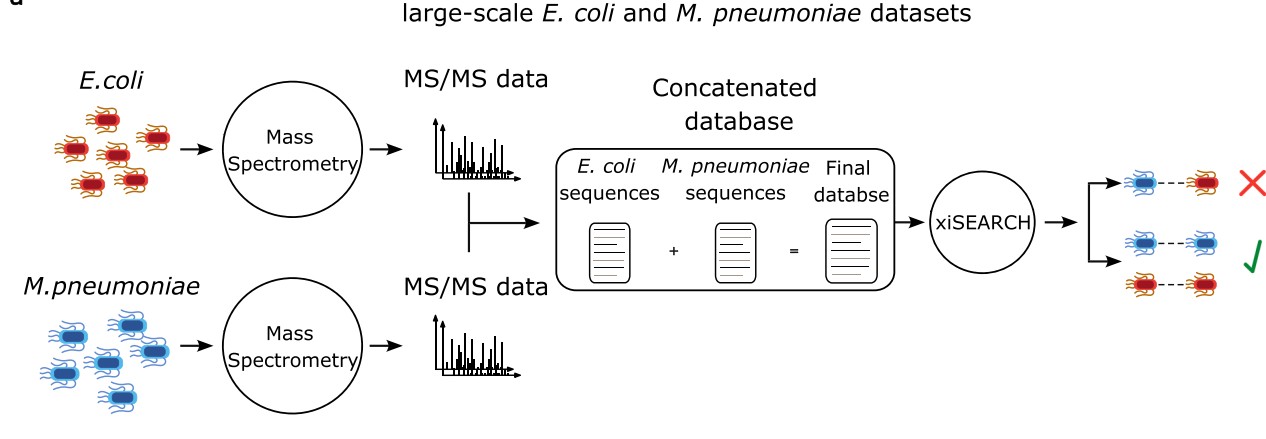

**b**

xiSEARCH+Prosit-XL+xiFDR vs. xiSEARCH+xiFDR
*E. coli* and *M. pneumoniae* datasets

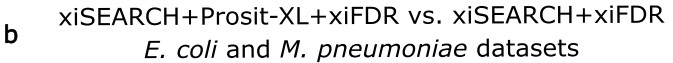

**c**

Comparison of target-decoy separation
with rescoring comparing to without

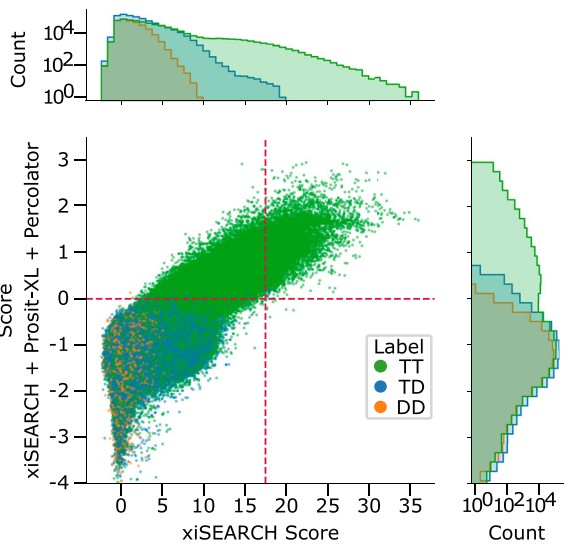

**Fig. 4 | Evaluation of Prosit-XL versus large-scale datasets and extensive search space. a** Schematic illustration of the experiment designed to estimate FDR in large datasets. Briefly, two distinct large-scale datasets (*E. coli* and *M. pneumoniae*) were analyzed together by xiSEARCH with a combined protein database. Any identified XL peptide suggesting a PPI between *E. coli* and *M. pneumoniae* is considered a false positive due to this being an organism mismatch. **b** Vennbars show the number of identified CSMs, peptide pairs, and PPIs lost (orange), shared (blue), and gained (green) at an FDR of 1% on CSM-, peptide pair-, and PPI-level when comparing results from xiSEARCH+Prosit-XL+xiFDR (second bars) to xiSEARCH

+xiFDR (first bars) on the results obtained from the experiment shown in **a**. Source data are provided in Supplementary Data 4. **c** Comparison of target-decoy separation on CSM-level using xiSEARCH scores (*x*-axis) and xiSEARCH+Prosit-XL +Percolator scores (*y*-axis). Green, blue, and orange dots represent individual target-target (TT), target-decoy (TD), and decoy-decoy (DD) CSMs, respectively. The marginal distributions show the respective score histograms. For illustration purposes, the *y*-axis of the marginal histograms is plotted in a log scale. The vertical and horizontal red lines indicate the 1% FDR cutoff applied at the CSM level, which yielded the results shown in **b**.

the actual FDR. This approach is similar to entrapment searches, but some number of false positives within-species matches will remain and cannot be accounted for in the estimate of the actual FDR of the analysis.

The improvement after rescoring was substantial, with the number of identified between-links increasing to ~5.6, ~5.7, and ~3.4 fold on CSM-, peptide pair-, and PPI-level, respectively, compared to applying only xiSEARCH+xiFDR (Fig. 4b). Despite this substantial increase in identifications, the number of mismatches remained very low: out of 2431 CSMs, only 13 mismatches (0.53%); out of 1671 peptide pairs, 6 mismatches (0.35%); and out of 517 PPIs, 4 mismatches (0.77%). This is because the features generated by Prosit-XL in combination with the Percolator-based CSM score generation approach can effectively

separate true positive target-target matches from false positive target-target matches due to an improved scoring of target-decoy (in part representing TP-FP) matches (Fig. 4c). As visible in the marginal histograms, the score distribution of target-decoy matches aligns much better with the false positive portion of the target-target matches when rescoring is used in comparison to xiSEARCH. This is because the Percolator-based CSM score takes the minimum of the individual PSM scores and thus a single incorrect peptide in an XL-peptide will lead to an overall poor score irrespective of whether the other peptide is in fact a true positive match. This leads to a shift in score cutoff necessary to achieve 1% FDR, resulting in a larger portion of matches surviving the FDR threshold. This experiment demonstrates the capability of rescoring pipeline compared to xiSEARCH in an extremely large search

space while maintaining control over the FDR. To the best of our knowledge, this result is the highest increase in identifications observed after CSM rescoring compared to previous studies. Further, we compared our rescoring pipeline to xiSEARCH by applying a lower bound estimate of the actual FDR of 1% (Methods). The results still show substantial improvement after rescoring, with the number of identified CSMs increasing from 1711 to 3389, peptide pairs from 1147 to 2378, and PPIs from 390 to 678 (Supplementary Fig. 5a).

Additionally, we examined the number of expected true positive matches by calculating #TT - (#TD - #DD) after the rescoring process for all analyzed datasets in this study (Supplementary Fig. 5b). The results clearly show that the estimated number of true positives is highest in the high-scoring region and, as expected for Percolator, drops to near zero around a score of 0. At lower scores, the number of estimated true positives remains around zero, indicating that the scoring approach and the use of machine learning did not introduce any unintended bias (i.e., artificially separating targets from decoys). This also suggests that the decoys (TD and DD) provide a reliable estimate for the number of false positive targets.

### Prosit-XL-assisted rescoring increases coverage and depth of 3D structure information and PPI mapping

In our final case study, we assessed our rescoring pipeline's ability to demonstrate its benefits in analyzing protein 3D structures and PPIs by applying it to a dataset that aimed to resolve the interactome map of intact human cytomegalovirus virions[42]. Briefly, we used xiSEARCH followed by the Prosit-XL assisted rescoring as described earlier and compared the results to original XlinkX analysis. The total number of unique interactions was 2427 using XlinkX (1% FDR at UXL level) and increased to 2910 with xiSEARCH+Prosit-XL+xiFDR (Fig. 5a). The biggest increases were observed at the human intra-protein level (1399 and 220 out of 2910), likely a result of the improved sensitivity at which UXLs can be detected by our pipeline. At the UXL-level, rescoring showed an almost 1.5-fold increase (from 4789 to 7396 UXLs) over XlinkX on human intra-protein-protein connection (self-links) (Supplementary Fig. 6a). Similarly, our rescoring pipeline was compared to xiSEARCH+xiFDR results. The most notable increase was observed for the inter-protein-protein interactions with around 3-fold increase (from 424 to 1203 inter-PPIs, Supplementary Fig. 6b). In addition, this increase was also seen with a ~2.3-fold increase at the UXLs level (from 2396 to 5460 UXLs for inter-PPIs, Supplementary Fig. 6b). Both xiSEARCH+xiFDR and xiSEARCH+Prosit-XL+xiFDR applied FDR in a more conservative manner at the CSM-, peptide pair-, and PPI-level.

To further investigate the gains and losses, we assessed the PPIs (inter- and intra-PPI) gained, lost, and shared between xiSEARCH +Prosit-XL+xiFDR and XlinkX for each category (Supplementary Fig. 6c). In general, rescoring added 548 new PPIs that were not detected by XlinkX. However, we also observed a loss of 351 PPIs (Fig. 5b, upper left Venn diagram), which are supported by 403 UXLs that our pipeline did not identify. Further, we investigated the gained, shared, and lost UXLs of PPIs shared between our pipeline and XlinkX. While rescoring led to a gain of 2,235 UXLs, we also observed a loss of 1044 UXLs (Fig. 5b, upper right Venn diagram). Despite the improvements in identified PPIs and UXLs, the number of lost UXLs (total of 1447) is rather high compared to the losses observed in the earlier analysis. To investigate the reason, we checked if the UXLs identified uniquely by XlinkX appeared in the unfiltered search results of xiSEARCH, revealing that ~89% of these UXLs were absent in it (Fig. 5b, bottom Venn diagram), indicating that only 155 were lost because they did not survive our conservative FDR cutoffs at all levels.

The increase in UXLs leads to recovering more PPIs at various cutoffs of minimum UXLs required to call a PPI, commonly applied to remove one-hit-wonders. This increase ranges from 30% for PPIs with at least 1 UXL to 37.5% for PPIs that are supported by at least 10 UXLs

(Supplementary Fig. 6d). This is confirmed by the observation that on average a PPI is supported by 1.32 more UXLs (linear regression model $y = 1.32x + 0.07$) using our rescoring workflow (Fig. 5c). As a result, the number of interaction partners identified for each protein, e.g., UL32, UL25, and UL83, increases (Supplementary Fig. 6e, scatter plot). UL83 and UL25 are major tegument proteins in HCMV and play essential roles in viral assembly. UL83 is crucial for tegument formation, where it helps stabilize the virion structure[43]. UL25, on the other hand, serves as a hub for assembling other viral proteins into the maturing virion, making it an organizing center during the virion maturation process[44]. It is reported that UL83 facilitates the incorporation of UL25 into mature viral particles[45]. The interaction between UL25 and UL83 is supported by the results, which show that 62 UXLs were identified for this interaction using rescoring, and 44 UXLs using XlinkX (Supplementary Fig. 6e, Table). Additionally, it is suggested that host proteins such as Grb2 and DDX3 show dependency on UL83, being incorporated into virions upon viral infection[46]. However, DDX3X's direct interaction with UL83 remains unclear. It is also worth mentioning that HCMV infection enhances the expression of Grb2 and DDX3X, facilitating viral replication and spread. As a result, DDX3X has emerged as a target for antiviral therapies due to its critical role in infection[46,47]. In our results, we observed an increase in UXLs for the UL83-DDX3X interaction; specifically, we found 34 UXLs using rescoring, while XLinkX shows 15 UXLs (Fig. 5d). Similarly, our results suggest the interaction between UL25 and DDX3X by enhancing the previously reported UXLs from 8 for XLinkX to 21 for rescoring, revealing a larger interaction area (Fig. 5d). These findings suggest that there may be a more complex relationship between UL83-UL25-DDX3X.

From the result of UXLs, the dataset shows a substantial number of self-links. We evaluated these UXLs separately for viral and human self-links. In the original study, the viral UXLs were evaluated by using a specific example for self-links of the protein UL55[42]. As it is given in the dataset paper, the distance between cross-linked residues should be under 40 Å[42]. We further looked into this protein and whether our results show an improvement in detecting UXLs based on this rule of thumb. Specifically, our analysis identified 44 unique UXLs (36 were reported in the XlinkX analysis), with 9 exhibiting distances greater than 40 Å in the post-fusion structure. Notably, all these UXLs adhered to the acceptable threshold in the pre-fusion structure (Fig. 5e). The term "fusion" here refers to the merging of the viral and host cell membranes mediated by glycoproteins, a critical step in herpesvirus infection. When comparing post-fusion to pre-fusion data, our findings align with the original paper (Supplementary Note 2).

To evaluate the human self-link UXLs, we define two metrics, including link distance and average plDDT (predicted local distance difference test). Link distance was calculated as the Euclidean distance between cross-linked K α-carbon using the protein structures retrieved from the EBI-AlphaFold2 (AF2) database[45]. The average plDDT was calculated as the average AF2 plDDT local confidence value for the linked LYS residues as one metric for each UXL. It is known that the plDDT value above 70 in the AF2 is defined as "high confident" prediction, and it performs good for the prediction of protein backbone structure[45]. Thus, we categorized our results into four distinct groups based on specific thresholds for distance (40 Å) and Av. plDDT (70) (Supplementary Fig. 6f). Our analysis demonstrated that the majority (~86%) of the UXLs were observed at a distance of <40 Å. Moreover, the highest density distributions were detected for distance <40 Å and average plDDT > 70, indicating that most detected UXLs resided within the 'high confidence' predicted regions, with sufficient distance to establish a linkage. Noteworthy, the number of <40 Å detected interactions with average plDDT <70 indicates that the plDDT estimate of AF2 may not be as well calibrated as expected and underpins the value of orthogonal information provided by XL-MS for resolving protein structures[48].

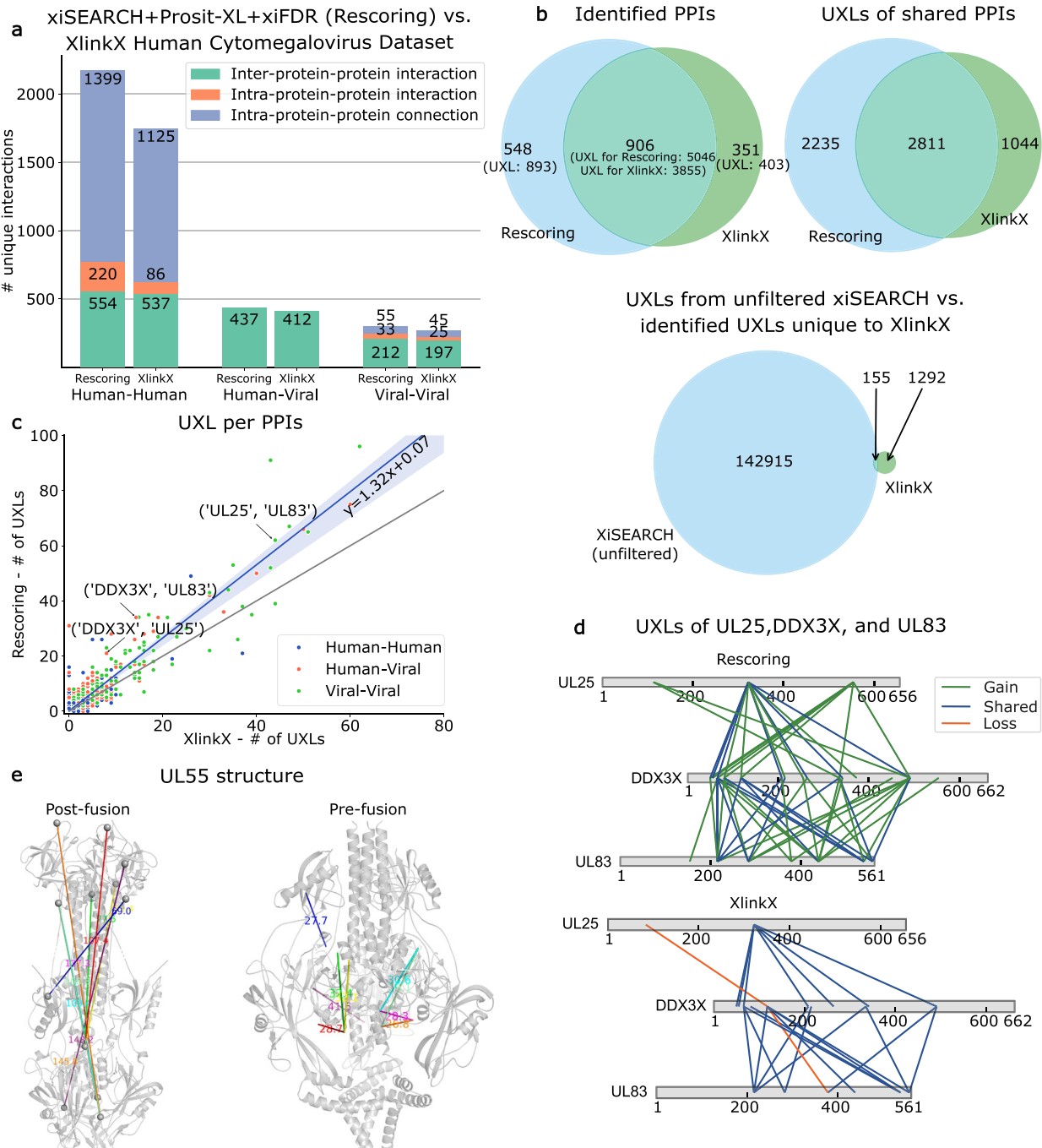

**Fig. 5 | Evaluation of Prosit-XL for analyzing 3D protein structures and protein-protein interactions. a** Unique number of identified interactions; inter-protein-protein interaction (green), intra-protein-protein interaction (orange), and intra-protein-protein connection (dark blue). The bars represent the number of PPIs and self-links for human-human, human-viral, and viral-viral interactions identified by data-driven rescoring of xiSEARCH results (left) and as reported in the original study by XlinkX (right). Source data are provided in Supplementary Data 5. **b** Venn diagrams comparing shared PPIs and UXLs between the rescoring results and the XlinkX results. The top left Venn diagram shows the PPI-level comparison for all types of PPIs. In parentheses, the total number of UXLs detected for the corresponding PPIs is shown. Top right Venn shows the UXL-level comparison for UXLs extracted from shared PPIs only (intersection in top left Venn). The bottom Venn diagram compares the UXLs in the unfiltered search results from xiSEARCH with the UXLs that were uniquely identified by XlinkX (combination of unique PPIs and missed UXLs from shared PPIs). **c** Correlation between UXL counts per PPI by rescoring versus XlinkX, separated by interaction type for human-human (blue),

human-viral (orange), and viral-viral (green). The diagonal and regression lines are shown in dashed black and solid blue, respectively. The shaded blue area around the regression line indicates the 95% confidence interval of the regression. Data are presented as mean ± SEM for the shared set of PPIs identified by both methods: rescoring shows a mean of 4.54 ± 0.33 UXLs per PPI, and XlinkX shows a mean of 3.52 ± 0.25 UXLs per PPI. **d** Network representations of the interactions among the viral proteins UL25, and UL83, as well as the human protein DDX3X, comparing results from rescoring and XlinkX. Solid green, blue, and orange lines present confidently identified UXLs by rescoring or XlinkX that were gained, shared, or lost when comparing rescoring to XlinkX, respectively. The network was visualized using xiView[54] and modified. **e** Structural representations of the post-fusion (PDB: 7KDD) and the pre-fusion (PDB 7KDP) conformations of the viral proteins UL55. Lines indicate confidently identified UXLs by rescoring, highlighting the distance of the interacting sites. Colors represent individual interactions mapped onto the structure. The residue alpha carbons were depicted as gray spheres. The UXL distances shown were calculated using PyMOL.

## Discussion

In this study, we introduce Prosit-XL, a deep learning model that can predict the fragment ion intensities of XL-peptides with high accuracy. We evaluated it on various experimental holdout sets, demonstrating its strong generalizability to new, unseen data. Further, Prosit-XL was integrated into a data-driven rescoring pipeline, to our best knowledge, the only rescoring pipeline that includes CSM rescoring. For this, an approach was developed that is splitting CSMs into PSM-level to allow the effective combination of our intensity-based features into a single score by percolator. The rescoring pipeline was analyzed using synthetic ground truth datasets, providing insight into its FDR estimation performance. Finally, the pipeline was applied to real-world datasets, resulting in substantial gains on CSM, UXL, and PPI levels, showing its utility and benefits in assisting protein 3D structure elucidation and interactome mapping at organism scale.

There has been notable progress in generating ground truth datasets in the XL-MS field, which is a positive step forward for benchmarking methods. However, small and less complex datasets, such as the analyzed synthetic peptide dataset, can pose challenges for benchmarking machine learning–based tools. These datasets may lack the necessary complexity to effectively benchmark tools for FDR calibration. Future work would benefit from incorporating larger and more complex datasets to enable more reliable benchmarking and further advance the field.

An important consideration when using post-processing tools such as Percolator is the potential for overfitting when the same data is used for both model training and scoring. While earlier versions of Percolator addressed this with cross-validation, this strategy did not fully prevent information leakage. The more recent RESET[49] approach improved upon this. While this strategy focuses on avoiding data leakage from decoys, all targets are still present during the training and scoring steps, and although our empirical results did not indicate any noticeable bias, the possibility of (subtle) residual biases cannot be entirely excluded. This remains an area where further methodological improvements could enhance the robustness of post-processing workflows, specifically for cross-linking.

Despite Prosit-XL's high accuracy, several potential improvements, beyond the scope of this current work, could enhance its performance even further. One potential improvement is incorporating separate charge information for each peptide, instead of the current use of one charge for the entire XL-peptide precursor. Current XL-DBSEs (e.g., pLink 2) do not provide accurate individual peptide charges due to the complexity of separating charges in XL-peptides. This is a burden for the model since it has to learn how to correctly estimate the charge state of each peptide, in addition to predicting fragment ion intensities. Additionally, MS2 spectra in crosslinking experiments are often acquired using stepped CE, where multiple CEs are applied in a stepwise manner during fragmentation. This can negatively impact the NCE calibration process, as finding a single optimal CE per MS file for Prosit-XL may not be ideal for capturing the effects of the multiple CEs used in practice. More research is required to estimate the importance of stepped CE in XL-peptide fragmentation and to develop corresponding calibration methods. However, a lack of ground truth systematic data does not allow a comprehensive investigation at this moment.

Although not demonstrated in this work, there is no technical reason why Prosit-XL could not be adapted to other spectra or other peptide properties, such as retention time. Potential improvement to further extend Prosit-XL's ability to predict spectra may be to accommodate other fragmentation techniques such as MS2-CID and MS2-electron-transfer dissociation[50] (ETD or EThcD), different fragment ion types (e.g., neutral losses), additional cross-linkers (e.g., DHSO[51], DMTMM[51]), and different cross-linked residues[1] (e.g., S, T, Y). Extending the model's capabilities in these areas could lead to even more comprehensive and accurate predictions.

As highlighted by our analysis, another future improvement in analyzing XL-MS datasets is the application of rescoring on search results from multiple XL-DBSEs, as losses observed when using different search engines may be attributed to differences in processing, rather than differences in confidence estimation. Although XL-DBSEs often identify overlapping sets of XL-peptides and PPIs, there are still cases that are uniquely identified by each XL-DBSE. Because our rescoring approach projects any CSM from any XL-DBSE to the same feature space, alternative explanations from different search engines can be compared, and the best can be picked. Taking advantage of this could lead to further substantial gains, increasing depth and coverage of XL experiments. Likewise, previous work in localizing PTMs has shown that true positive modified peptides may not be present as the highest scoring match produced by a DBSE. Because Prosit-XL takes both peptides into account for prediction, in combination with data-driven rescoring, a better ranking of XL-peptides may yield further gains in the future.

Overall, Prosit-XL represents a valuable advancement in XL-MS, specifically also due to its direct integration into Koina and Oktoberfest, enabling any scientist to benefit from deep-learning-assisted data analysis. Its ability to provide intensity predictions for complex datasets and enhance the identification of XL-peptides and PPIs through rescoring makes it a powerful tool for studying protein structures and protein-protein interactions, particularly for in vivo studies, as exemplified by the exceptional performance on the two-species mix dataset. As the field continues to evolve, and MS focusing on linear peptides already strongly relies on high-quality predictions, we are convinced that Prosit-XL's capabilities and integration into rescoring will be key for advancing XL-MS and thus our understanding of protein interactions in vivo at proteome-scale.

## Methods

### Training data

In the process of data collection, we used 11 publicly available datasets: (1) DSBU: PXD012546; (2) DSSO: PXD019926, PXD017711, PXD011861; (3) DSS/BS3: PXD017620, PXD016554, PXD019926, PXD017695, PXD014675, PXD008550. MS2 spectra were acquired by HCD fragmentation, followed by analysis using the Orbitrap with high resolution and mass accuracy, except PXD019926, which also contains CID MS3 spectra. MS2 spectra were searched by pLink 2[7], a high-speed search engine for proteome-scale identification of XL-peptides, and hence is very suitable for generating large-scale XL benchmark datasets, which are valuable for deep learning models. pLink 2 was used with the following parameters: Carbamidomethylation on cysteine and oxidation on methionine as fixed and variable modifications, respectively; peptide masses ranging from 600 to 6000 Da; precursor and fragment mass search tolerance set to 10 and 20 ppm; maximum allowed missed cleavages set to 3; crosslink specificities at lysine residues and protein N-terminals, FDR set at 0.5% at the CSM level. The extracted spectra were further filtered out under the following conditions: peptide lengths (peptides A and B) less than 6 or greater than 30, precursor charge greater than 6, and the number of matched peaks less than the length of each peptide. Ultimately, the top 10 CSMs for each unique XL-peptide were kept to avoid too much redundancy. All extracted spectra were annotated, where all expected b, y, b-xl, and y-xl for NMS2 spectra and b, y, b-short, y-short, b-long, and y-long, with for CMS2 spectra, charges up to 2 for CMS2 and 3 for NMS2 spectra, are calculated and matched against experimentally acquired fragment peaks. Matching tolerances were 20 ppm for FTMS. Ultimately, the annotated spectra were split into three distinct sets: training (80%), validation (10%), and holdout set (10%). To minimize data leakage, if an XL-peptide pair A-B is included in the training set, neither peptide A nor peptide B can be in the validation or holdout set. Regarding NCE, all MS files were analyzed by MS Amanda[52] with default parameters to identify linear

peptides, which are then used for optimal NCE estimation. To clarify, the top 1000 PSMs were chosen based on MS Amanda's score and were compared to predicted spectra by HCD Prosit 2020 at different NCEs ranging from 18 to 49. The optimal NCE was determined by identifying the NCE at which the highest SA is observed.

## Input and output of Prosit-XL

Inputs to the model are peptide sequence A, peptide sequence B, NCE, and precursor charge. Peptide sequences are encoded as integer vectors of length 30, with each integer representing a specific amino acid, and fed to an embedding layer. For cross-linked lysine residues, a unique integer is assigned to indicate the crosslinker, depending on the crosslinker type. Sequences shorter than 30 amino acids are padded with zeros. The precursor charge is represented using one-hot encoding. The Prosit-XL's output is annotated spectra, which are transformed to a tensor. Ion intensities are normalized continuous values. A CMS2 and NMS2 spectrum are represented by a 348-dimensional vector (y/b/ys/bs/yl/bl ions, 3 charges, 29 fragment ions) and 174-dimensional vector (y/b/yxl/bxl ions, 3 charges, 29 fragment ions), respectively, and orders as follows: y1 (1+), y1 (2+), y1 (3+), b1 (1+), b1 (2+), b1 (3+), y2 (1+) and so on. The type of fragment ion can change based on the position of the crosslinker. For example, if the crosslinker is attached to the first amino acid in a non-cleavable XL-peptide, b1 actually represents b-xl 1, indicating a modified b-ion.

## Prosit-XL architecture

*Encoder 1 and 2:* The encoder 1 and 2 contain an embedding layer, followed by two bi-directional recurrent neural networks (BDN) with gated recurrent memory (GRU) units, connected to an attention layer. The recurrent layers use 512 memory cells each. *Latent space:* The latent space of each encoder is 512 units for each amino acid token. *Encoder 3:* Precursor charge and NCE encoder is a single dense layer followed by dropout. The latent vectors from Encoders 1 and 2 are first multiplied elementwise. The resulting product is then multiplied with the output of Encoder 3. *Decoders 1 and 2: Both decoders consist of* a one-layer length 29 BDN with GRUs. It is important to note that Decoder 2 is specifically developed for CMS2 spectra covering y-long and b-long fragments.

## Prosit-XL training process

We applied transfer learning using the HCD Prosit 2020 and CID Prosit 2020 weights as starting points and then trained these using the CMS2 and CMS3 training sets to develop Prosit-XL-CMS2 and Prosit-XL-CMS3, respectively. The model weights of Prosit-XL-CMS2 were used as the starting point for the development of Prosit-XL-NMS2 using the NMS2 training set. To control for overfitting, early stopping was employed on the validation set scores, employing a patience of 20 epochs. The holdout set was used after the model was fully trained to evaluate its generalization and potential biases. The loss function was the normalized spectral contrast loss. We used the Adam optimizer with a cyclic learning rate algorithm. During training, the learning rate cycled between a constant lower limit of 0.00001 and an upper limit of 0.0002, which is continuously scaled by a factor of 0.95 with the "triangular" mode. The model was trained with a batch size of 2000[36].

## Prosit-XL's performance on synthetic peptide datasets

**Synthetic dataset cross-linked by DSSO.** All 3 MS files (1, 2, and 3 replicate) were downloaded from the PRIDE repository with the identifier PXD029252. MS files were searched using xiSEARCH with the following parameters: report_top_ranking_only: false, delta_score_filter: false, enzymes: trypsin, missed_cleavages: 2, min_peptide_length: 6, max_peptide_length: 30, isotope_error_ximpa: 2, noncovalent_peptides: true, threads: 20, ms1_tol: 10 ppm, ms2_tol: 10 ppm, top_n_alpha_scores: 10, top_n_alpha_beta_scores: 10, crosslinker: {name: "DSSO", mass: 158.0038, specificity: K}, conservative_n_multi_loss: 3, denoise_alpha:

{top_n: 10, bin_size: 100}, denoise_alpha_beta: {top_n: 20, bin_size: 100}, fragmentation: {nterm_ions: b, cterm_ions: y, add_precursor: true, max_nloss: 4, match_missing_monoisotopic: true}, max_var_protein_mods: 2, max_modified_peps: 20, modification1: {name: cm, specificity: C, type: fixed, composition": "C2H3N1O1"}, modification2: {name: ox, specificity: M, type: variable, composition: O1. Next, xiFDR was applied with the following parameters: the FDR level for CSM, peptide pair, residua pairs, and protein pairs was set to 1%, without boosting. The identified CSMs were verified against provided groups to remove potential identified false-positive TTs. Finally, the final list of identified CSMs was submitted to Prosit-XL-CMS2. The NCE calibration was performed by Prosit-XL-CMS2. The SAs were calculated separately for peptide A and B (Fig. 2c).

**Synthetic dataset cross-linked by DSS.** MS file was downloaded from the PRIDE repository with the identifier PXD014337 and analyzed by pLink[53] according to the following parameters: Crosslink mass: 138.068, monolink mass: 156.079, crosslinker reactivity: K-K, fixed modification: Carbamidomethyl, variable modification: Oxidation, enzyme: trypsin, max. Missed cleavages: 3, Min peptide mass: 500, Max peptide mass: 6000, Min peptide length: 5, Max peptide length: 60, MS1 tolerance (ppm): 5, MS2 tolerance (ppm): 20, FDR: 1% at PSM level. The identified CSMs were verified against provided groups to remove potential identified false-positive TTs. Subsequently, we applied extra filtering and removed CSMs with scores less than 0.03 and applied Prosit-XL-NMS2. The NCE calibration was performed by Prosit-XL-NMS2. The SAs were calculated separately for peptide A and B (Fig. 2c).

## Comparison of Prosit-XL and pDeepXL's performance on synthetic peptide datasets

All identified CSMs described in the previous section (Fig. 2c) were submitted to pDeepXL for prediction. Some CSMs were removed due to DeepXL's limitations, such as restrictions on peptide length. For predicting MS/MS spectra of the synthetic peptide dataset linked by DSSO, the following pDeepXL parameters were used: instrument: QEHF, NCE_low: 21, NCE_medium: 27, NCE_high: 33, and crosslinker: DSSO. For the dataset linked by DSS, the parameters were: instrument: QEHFX, NCE_low: 0, NCE_medium: 28, NCE_high: 0, and crosslinker: DSS. It should be emphasized that SAs and PCCs are measured separately for peptide A and peptide B, and only for CSMs that both pDeepXL and Prosit-XL could predict (Fig. S2d).

## General rescoring pipeline

The rescoring pipeline (Fig. 3a) requires MS2 spectra files, either in RAW or mzML format, and unfiltered XL-DBSE's output (xiSEARCH or Scout) as inputs, which contains both target (TTs) and decoy (TDs/DDs) CSMs. Annotation of MS2 spectra is then performed by calculating all potential b- and y-ions for CSM2 spectra (b, y, b-short, y-short, b-long, y-long) and NMS2 (b, y, b-xl, y-xl) with charge up to 2 and 3 for CMS2 and NMS2, respectively. These potential fragments are matched against the experimentally acquired fragment peaks with a 20 ppm mass tolerance for FTMS. Next, optimal NCEs are determined by calibrating Prosit-XL to each provided MS file. Specifically, the 20 highest-scoring CSMs are selected, and then NCE as Prosit-XL's input is adjusted in a reasonable range (18 to 49). The NCE that leads to the highest SA between predicted and acquired spectra is used as Prosit-XL's input for that MS file. With the prediction from Prosit-XL at an optimal NCE, Oktoberfest then generates ~150 features per CSM (separately calculated for peptide A and B). The list of features with their corresponding descriptions is provided in Supplementary Table 1. Rescoring is performed on the PSM-level, where each peptide with its corresponding ~75 features is submitted to Percolator (v 3.6.1). Percolator is only used to aggregate the features into a single score, and no q-value or other FDR estimate is taken from it. The final CSM-score is constructed by taking the minimum PSM-level percolator

discriminant for each CSM. Additionally, rescoring was also performed using the latest version of Percolator (v3.7.1), with the results presented in Supplementary Fig. 7. It is crucial to highlight that the order of peptides A and B in the xiSEARCH or Scout output does not affect the Prosit-XL's performance, the features generated by Oktoberfest, and the final Percolator scores. Ultimately, CSM, peptide pair, and PPI level FDR were estimated using xiFDR[3], where all FDR levels (PSM, peptide pair, residual pairs, and protein pairs) are set to 1% without boosting. FDR calculations for self and between links are done separately using FDR = (TD-DD) / TT. It is important to note that CSMs provided by xiFDR are unique CSMs, meaning for any given peptide pair, modifications, link sites, and charge state combination only the top scoring one is reported. More information about xiFDR can be found at https://github.com/Rappsilber-Laboratory/xiFDR.

### Application of Prosit-XL and rescoring
**Rescoring of synthetic peptide dataset cross-linked by DSSO.** After running xiSEARCH on 3 MS files, rescoring was applied to each replicate separately, with all features in Oktoberfest config file set to false. The number of identified CSMs and peptide pairs (both self- and between-links), by xiSEARCH+xiFDR and xiSEARCH+Prosit-XL+xiFDR, shown in Fig. 3b, represents the average of CSMs and peptide pairs per replicate. The experimentally validated FDR is calculated using the following formula: FDR = (TTs not within the same XL group) / (TTs total) for each replicate. The experimentally validated FDR in Fig. 3b shows the average actual FDR per replicate.

**Applying an actual FDR of 1% to the synthetic peptide dataset cross-linked by DSSO.** All CSMs provided by Oktoberfest were submitted to xiFDR with an initial FDR of 100%. Next, for the files generated by xiFDR at the CSM and peptide pair levels, all TDs and DDs were removed. The remaining TTs CSMs, and peptide pairs were then sorted once based on the Percolator score and once based on the xiSEARCH score. Finally, an actual FDR of 1% was applied using the formula: FDR = (TTs not within the same XL group)/(total TTs).

**Rescoring of synthetic protein dataset cross-linked by DSSO.** MS files were kindly shared with us by the Liu lab and were analyzed using Scout (v.1.4.14) by the following parameters: add contaminants = false; add decoys = true; fragment bin tolerance = 0.02; fragment bin offset = 0; max fragment bin m/z = 1800; min fragment bin $m/z$ = 200, deconvolution for MS searching = false; deconvolution for ion pair = true, crosslinker = DSSO; target n-term = true; reaction residuals = k; enzyme = trypsin; enzyme specificity = FullySpecific; Isotopic possibilities precursor = 1; min peptide length = 6; max peptide length = 60, max variable modification per peptide = 2; min peptide mass = 500, max peptide mass = 6000, max miscleavages = 3, ppm error on MS1 level = 10, ppm error on MS2 level = 20, static modification = Carbamidomethyl, variable modification = Oxidation on Methionine; FDR 100% on all level. The final Scout score (Classification score) was used as input for xiFDR (all levels set to 1% without boosting). After rescoring the unfiltered result of Scout, the Percolator score is used as xiFDR input. The actual FDR is calculated by FDR = (FPs)/(FPs + TPs). For TP identification, proteins need to be in the same group and batch. Notably, only identified between-links are presented in Fig. 3c. For more details, see ref.[10].

**Rescoring of large-scale dataset cross-linked by DSSO.** Two distinct subsets of large-scale datasets containing *E. coli* (JPST000845) and *M. pneumoniae* (PXD017711) were analyzed together using xiSEARCH. The analysis used the same parameters as those used for the synthetic XL-peptide dataset. To control for FDR estimation, a combined database search including both *E. coli* and *M. pneumoniae* protein sequences was performed. In this combined search, any proposed interaction between *E. coli* and *M. pneumoniae* protein sequences identified by a

CMS/UXL was considered a FP identification, labeled as mismatch. Similar to the synthetic XL-peptide dataset, xiFDR is used twice, separately for inputs of the xiSEARCH' score and the percolator score, after the rescoring process. It is important to note that only identified between-links are shown in Fig. 4b.

**Applying a lower bound estimate of the actual FDR of 1% to the large-scale dataset cross-linked by DSSO.** All CSMs provided by Oktoberfest were submitted to xiFDR with an initial FDR of 100%. Next, for the files generated by xiFDR at the CSM, peptide pair, and PPI levels, all TDs and DDs were removed. The remaining TTs, CSMs, peptide pairs, and PPIs were then sorted once based on the Percolator score and once based on the xiSEARCH score. Finally, a lower bound estimate of the actual FDR of 1% was applied.

**Rescoring of human cytomegalovirus dataset cross-linked by DSSO.** The dataset with the identifier PXD031911 was analyzed by xiSEARCH with the same parameters as those used for the synthetic dataset (linked by DSSO). Our evaluation of Prosit-XL's performance involved comparing UXLs identified at 1% CSM-, peptide pair-, and PPI-level FDR by xiSEARCH+Prosit-XL+xiFDR and those identified at 1% UXL-level FDR by XlinkX. Comparison is done via two levels: PPIs and UXLs. PPIs were defined as a combination of inter-protein-protein interactions where both proteins are different and intra-protein-protein interactions where there is a sequence overlap on the peptide pair. Same definitions and approaches were applied for xiSEARCH+xiFDR search results. For PPI level comparison, the gene name of the UXL is taken and sorted to unify naming coming from both xiSEARCH+Prosit-XL+xiFDR and XlinkX (example PPI; ("UL25", "UL83")). For the UXL level, protein link position is included for respective proteins (example UXL; ("UL25", "84"), ("UL83", "557")). To tackle the complexities of ambiguous UXLs, we implemented a standardized selection process. The dataset paper previously provided results from XlinkX along with a selection of UXL for ambiguities[42]. Initially, we utilized the XlinkX results to determine which proteins were involved in ambiguous UXLs. We then aligned the UXLs from xiSEARCH+Prosit-XL+xiFDR with those obtained from XlinkX. Following this alignment, we selected the UXL based on those provided by XlinkX for the corresponding matched UXL. In cases where multiple UXLs were possible but did not align between the two methods, we chose the first candidate based on an alphabetically sorted list for xiSEARCH+Prosit-XL +xiFDR results. An initial comparative analysis was performed at the PPI level. In this context, PPIs were categorized as 'gain' if they appeared only in the results from xiSEARCH+Prosit-XL+xiFDR, "shared" if they were found in both xiSEARCH+Prosit-XL+xiFDR and XlinkX results, and "loss" if they were present only in XlinkX results. A subsequent comparative analysis focused exclusively on the "shared" PPIs. We extracted all UXLs from these shared PPIs for comparison. In the final analysis, we combined the UXLs identified for the "loss" PPIs from the first analysis with the UXLs defined for "loss" UXLs (those appearing only in XlinkX) from the second analysis. These combined UXLs were then compared against the unfiltered results from xiSEARCH. Additionally, we explored all combinations of "Human" and "Viral" interaction types (Human-Human, Human-Viral, and Viral-Viral) while utilizing all PPIs and UXLs from xiSEARCH+Prosit-XL+xiFDR and XlinkX. The resulting findings were visualized using a Venn diagram (see Fig. 5b and Supplementary Fig. 6a, b). Regarding AF2-based UXL distance measurement, the human protein structures were extracted from the EBI-AF2 database with reference proteome UP000005640. This database contains 23,391 predicted structures. With these settings, we were able to calculate 97% of the defined self-link for Human proteins. For the Human-Human self-link UXL distance measurement, we utilized the Euclidean distance and atomic coordinates. The UXLs' atomic coordinates were obtained using BioPython PDB or CIF file parsers. As all AF2 predictions only provide chain A, we measured the distance from only one chain. The distance measurements for UL55 were conducted using post-fusion (PDB: 7KDD)

and pre-fusion (PDB 7KDP) conformations. Since the UL55 structure is a trimer, UXL interactions can potentially take place between any of the chains. Consequently, distance measurements were conducted for UXL interactions located in all possible chain combinations. If the measurements for all the combinations show a distance of more than 40 Å, we re-calculated the same pairs using opposite structural conformations to validate the suitability of the UXL. Subsequently, these UXL interactions were manually evaluated using PyMOL.

## Reporting summary
Further information on research design is available in the Nature Portfolio Reporting Summary linked to this article.

## Data availability
The MS files containing CMS2 spectra of training, validation, and holdout set are available via the PRIDE repositories with the identifier PXD012546, PXD017711, PXD019926, and PXD011861. For the NMS2 spectra, the relevant identifiers are PXD017620, PXD016554, PXD019926, PXD017695, PXD014675, and PXD008550. The MS files for CMS3 spectra can be found under the identifier PXD019926. The training, validation, and holdout sets are available on Zenodo. The triton-compatible format of Prosit-XL-CMS2, Prosit-XL-CMS3, and Prosit-XL-NMS2 has been deposited on Zenodo and can be downloaded via the following links: https://zenodo.org/records/10277646, https://zenodo.org/records/10281001, and https://zenodo.org/records/11259344, respectively. The MS files for synthetic peptide datasets using DSSO and DSS cross-linkers can be found under the identifiers PXD029252 and PXD014337, respectively, while the synthetic protein dataset is available with identifier PXD042173. Additionally, the *E. coli* and *M. pneumoniae* dataset was sourced from JPOST identifier JPST000845 and PRIDE identifier PXD017711, respectively. The MS files of the human cytomegalovirus dataset were downloaded from PRIDE with the identifier PXD031911. The unfiltered search engine results.fasta files, rescoring results, and RAW files of human cytomegalovirus dataset have been deposited in the PRIDE repository with the identity of PXD057705. Regarding supplementary data files, the name of MS files used for data collection (training, validation, and holdout set) as well as those used in the rescoring process for each dataset are listed in Supplementary Data 1. The output of xiFDR (identified CSMs, peptide pairs, and PPIs) for the synthetic peptide dataset, synthetic protein dataset, *E. coli* and *M. pneumoniae* dataset, and human cytomegalovirus dataset is available in Supplementary Data 2–5, respectively. Source data are provided with this paper.

## Code availability
Source code and scripts are available on GitHub at https://github.com/wilhelm-lab/koina, and https://github.com/wilhelm-lab/oktoberfest. Oktoberfest repository is released under the MIT License, with all original license and copyright information retained. Attribution to reused components and dependencies is provided within the repository. Custom scripts for data analysis (e.g., cross-link distance calculation) were implemented in python and are available upon request.

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

## Acknowledgements

The authors acknowledge all members of the Wilhelmlab, as well as Manuel Matzinger, Micha Johannes Birklbauer, and Tara Bartolec, for their valuable input and discussions. We also thank the Liu Lab, especially Prof. Dr. Fan Liu and Max Ruwolt, for input and sharing their data regarding the human cytomegalovirus dataset. Additionally, we also thank Si-Min He's lab, particularly Prof. Dr. Si-Min He and Zhenlin Chen, for providing the pLink 2 search results. This work was in part funded by the European Union's Horizon 2020 research and innovation program under the Marie Skłodowska-Curie grant agreement (Grant No. 956148), an ERC Starting Grant (Grant No. 101077037), and German Federal Ministry of Education and Research (BMBF) (Grant No. 031L0305A).

## Author contributions

M.W. and J.R. jointly supervised the research, providing guidance throughout the project. M.K., C.S., L.F., and F.S. analyzed the data. M.K. trained and evaluated Prosit-XL models. M.K. and M.P. extended XL rescoring in Oktoberfest. M.K., C.S., J.L., and M.W. wrote the manuscript. All authors reviewed and approved the manuscript.

## Funding

## Competing interests

M.W. is a founder and shareholder of MSAID GmbH with no operational role and member of the scientific advisory board of Momentum Biotechnologies. The remaining authors declare no competing interests.
