## [Transparent Peer Review file · Nature Communications]

Prosit-XL: enhanced cross-linked peptide identification by fragment intensity prediction to study protein interactions and structures

Corresponding Author: Professor Mathias Wilhelm

Version 0:

Reviewer comments:

Reviewer #1

(Remarks to the Author)

Kalhor et al. present Prosit-XL, an extension of the spectral prediction framework Prosit to cross-linked peptides. Prosit-XL is designed to support cleavable and non-cleavable reagents alike. To compensate for the somewhat limited datasets for training that are available (compared to linear peptides), a transfer learning strategy was implemented. Impressive results as judged by spectral angles and Pearson correlation were achieved for all data types. Prosit-XL was integrated into a rescoring approach using the Oktoberfest and Percolator pipelines and first applied to existing benchmark data sets, where it showed promising performance. Subsequently, spectral prediction and PSM rescoring was applied to more complex data sets to demonstrate that it can recover increased numbers of protein-protein interactions and higher cross-link coverage between interaction partners alike.

Taken together, Prosit-XL is a very promising enhancement to Prosit and might be the best performing approach for the prediction of MS/MS spectra in the cross-linking field reported so far. The selected applications presented in this manuscript already show the potential of spectral prediction and rescoring for cross-linking applications, where it should be even more beneficial than for unmodified, single-chain peptides because of poor fragmentation and/or poor spectral quality due to the low abundance of cross-linking products.

General comments:

Throughout the manuscript, the authors use results from several cross-link search engines, including pLink, XlinkX, Scout, and xiSearch. While this to some extent highlights the general applicability of their approach, it comes at a risk of convoluting the effects of search engine performance and rescoring performance, at least in some cases. FDR control strategies will also be different between the pipelines, and both pLink and XlinkX have shown large deviations between predicted and actual FDRs in several studies. As an example, the CMV dataset compares XlinkX with xiSearch + Prosit-XL. Why not re-search the data with xiSearch first without applying Prosit-XL?

In several examples, the numbers go up with respect to CSMs, PPIs etc., although the actual FDR appears to increase as well (for cases where ground truths or entrapment designs are available), see data reported in Figs. 3a and 4a. I feel it would make more sense to try to adjust the thresholds in a way to make the actual FDR comparable between the +/- Prosit-XL workflows, which should be feasible?

In the introduction, the authors mention pDeepXL as previous work on the topic. Is it possible to somehow compare the performance of pDeepXL with Prosit-XL? If not, please explain why not.

Although in the end it probably does not matter so much, how is the order of peptides A and B defined? Based on the search engine output, or based on a definition of Prosit-XL? In line 162, the authors state that "on average, peptide B is shorter ...". If the longer peptide is always designated as A/alpha as is commonly done, then this should be an obvious conclusion (unless the two peptides have the same length).

The figures are very dense, and the smaller text is hardly legible even at 200% zoom factor. One particular example is Fig. 4a, where I could find a typo ("Final databse") only after zooming in even further. Some of the SI figures are even worse in this regard.

Additional comments:

Line 41: A frequently given argument is that the n^2 problem / search space expansion "increases the chance of false positive identification". I would argue that as long as FDR is controlled properly, the increased search space will rather increase false negatives because more stringent score thresholds need to be applied. Should tools such as xiFDR not handle that appropriately?

Line 190: From this discussion, it appears that features were introduced to Oktoberfest that are specific to cross-linked peptides. If so, they should be discussed somewhere in the manuscript.

Line 207: Although I agree that the strategy to use the minimum score of the two peptides for assessment of quality is sensible, I do not think that Supplementary Information 1 provides a real justification or adds more to the discussion.

Line 440: b1 (2+) is listed twice, probably it should say (3+) in one case.

Line 861: Although this is a paper about AI, the proteins are probably still viral rather than virtual ...

SI1, line 6 of the paragraph: Why do target-target matches also comprise FP-FP matches?

(Remarks on code availability)

I have checked the availability/accessibility of the code on GitHub and additional data deposited to Zenodo, but I have not attempted to install any software/run any code.

I have checked the publications/preprints related to Koina and Oktoberfest.

Reviewer #2

(Remarks to the Author)

This paper describes a pipeline for carrying out database search on mass spectra derived from cross-linked peptides. The key innovation is to integrate into the scoring various features produced by a deep learning model that predicts fragment ion intensities, which yields a large boost in statistical power. Another claimed innovation is a way to do FDR control using Percolator in this setting. I like several features of this work. For example, the approach to use linear peptides to estimate effective collision energy for cross-linked peptides and then use the estimated CEs for calibration (p. 3) is sensible and seems to work well. Furthermore, the model design, and particularly how previous models were adapted for transfer learning (p. 4 and Figure 2a), is very elegant. Overall, the work appears to be significant, but I also have some significant concerns that are outlined below.

One concern is with respect to novelty. The manuscript mentions that there are several existing methods that use post-processors for MS-XL analysis (line 52), but there is no discussion of the relative merits of their approaches relative to the one outlined here. Indeed, I think the idea of scoring two PSMs separately and then taking the min, which is presented as a key idea here, is already part of Kojak. More generally, what seems to be missing is a description of related methods, pointing out their differences, especially with respect to Prosit-XL.

The second concern is with respect to the validity of the approach. One of the main claims here seems to be that the proposed pipeline carries out "accurate and precise FDR estimation." I have several critiques related to this claim.

First, in the context of linear peptide analysis, it has already been established that PSM-level FDR control is problematic, because the decoy peptides do not adequately account for "neighbor" peptide relationships in the target database, where those neighbors arise due to homology (<https://arxiv.org/abs/1501.00537>). As such, I think that all of the analysis here should be done at the (cross-linked) peptide level.

Second, and on a related note, I could not find any indication of what version of Percolator was used for this analysis. This is important not only for reproducibility, but because the recent version fixed a longstanding problem with Percolator's FDR control. An important feature of the current version of Percolator is that it does FDR control at the peptide, rather than PSM level.

Third, and perhaps most worryingly, the claim in the abstract that the proposed method does a good job of FDR control is belied by the results of your first experiment. On p. 6 we learn that, at a nominal FDR threshold of 1%, the proposed pipeline yields an error rate of 1.18% at the CSM level and 1.65% at the peptide pairs level. It is true that this is much better than the reported results for competing methods (reported later in the paragraph to be 2.2-5.7%). But it still does not constitute valid FDR control.

Fourth, more generally, I missed in the manuscript any theoretical argument for why the proposed method should actually

control the FDR. We are only provided with empirical results on several datasets, which is obviously insufficient to guarantee FDR control in general. There is a pretty convincing proof of FDR control in the cross-linked setting (Walzthoeni et al.), albeit one that does not incorporate a machine learning post-processor. I am surprised that none of this theory is applied in the current setting.

In general, I think it's unfortunate that you've chosen to adopt the term "actual FDR" (line 212). In the statistics literature, I think the false discovery proportion (FDP) is the accepted term here, where the FDR is the expectation of the FDP. You also switched from "actual FDR" to "real FDR" on line 249.

I was struck by the fact that, in the section on large-scale datasets, you talk about using a method that is "similar to entrapment searches." Did you consider using one of the established entrapment protocols. For example, there is a recent preprint (Wen et al. 2024) that outlines several such methods. Why not use one of those here?

Minor points:

line 52: Kojak uses Percolator, not PeptideProphet, as claimed here.

line 64: "indicating" should be something like "yielding"

line 71: You say that Prosit-XL "retained" its CE-awareness. I don't understand how it can "retain" anything, since Prosit-XL was introduced in this paper.

line 121: I think you should start a new paragraph here.

line 158: I found this long list of ten SA values awkward to parse. I would recommend just saying something like "improved by XX--XX%", since the specific values are reported in the accompanying figure.

line 170: "remarkable" -> "remarkably"

line 232: What is "FP"?

(Remarks on code availability)

Reviewer #3

(Remarks to the Author)

Kalhor et al. present a study that uses transfer learning to apply Prosit to the problem of intensity prediction for CSMs – crosslinked spectrum matches. These CSMs are collected from crosslinked peptides arising from a range of different crosslinking scenarios (synthetic crosslinked peptides to in situ crosslinking), and using different styles of crosslinker (cleavable and noncleavable). The success of intensity prediction is parlayed into an elegant rescoring approach that leads to demonstrable improvements in performance across a range of XL-MS problems.

The paper is excellent, and the authors are commended for making the underlying methodology clear and easy to understand. It is well written. Prosit-XL provides a considerable performance boost that the community will welcome.

I have only two noteworthy comments to make, and then a few very minor ones:

1. Some additional commentary should be provided around the surprisingly high CSM losses shown in figure 3b,c (and to a lesser degree in figure 4b). This isn't the same as the imperfect overlap in figure 5b, where it is expected that XlinkX and xiSEARCH won't generate the same results. But in figure 3 and 4, the comparison involves xiSEARCH with/without PrositXL. Are the losses because of misidentifications in the search that doesn't use PrositXL? That doesn't make sense though, given the highly controlled FDR. Basically, why are so many high quality CSMs dropped?

2. The authors' use of the term PPI is misleading in several locations. This comes up particularly in the treatment of the HCMV datasets, but I think other places as well (hard to tell). They define inter-protein crosslinks as "between-links" and intra-protein crosslinks as "self-links" but then declare PPIs to equal self + between links (line 277). This is not correct. Please use PPIs to represent only the inter-protein data and the clearly unambiguous intra-protein data (i.e., only where the linked peptides share sequence). All others should be assumed intra-protein linkages and thus not a PPI, just a protein. It is important to be consistent with other papers on this point.

Very minor matters:

1. Line 268ff: it is more correct to say you are maintaining control over the FDR, not the FDR calculation.
2. Line 345ff: underpins the value OF orthogonal information...
3. Line 496: spelling of xiSEARCH
4. Line 526: spelling of replicate

(Remarks on code availability)

Version 1:

Reviewer comments:

Reviewer #1

(Remarks to the Author)

In this revised version, the authors have addressed all my comments. Specifically, they performed additional data analysis as suggested, improved the clarity of the figures and provided additional Supplementary Information.

(Remarks on code availability)

Reviewer #2

(Remarks to the Author)

Thank you for clarifying that the FDR control is done by xiFDR rather than Percolator. You mentioned this in the initial manuscript, but I missed that important detail. Anyway, now that I understand the workflow, I still have a concern: don't you expect the SVM scores coming from Percolator to be biased because they derived from the same data that was used to train the model? This is what the original Percolator cross-validation strategy tried (and failed) to address and what the new "pseudo-target" approach solves. I am sceptical that any strategy that treats the machine learning model as a black box can correctly control the FDR, unless the model is trained on different data entirely (both target and decoy). Supplementary Figure 4 does seem to provide empirical evidence that such a bias is not happening, but a priori one would expect the bias to be there, so it seems like some explanation is in order for why it's not happening. Relying on the fact that your ML model has relatively low capacity and strong regularization seems like a risky strategy.

I'm not really sure what to suggest at this point, since you have empirical results in hand that are consistent with reasonable (albeit imperfect) FDR control. Perhaps the most straightforward would be to add text to the discussion that addresses these uncertainties (as well as the weird results in Supplementary Figure 3 and 4 for the synthetic data).

At line 224, I still think you should not refer to "experimentally validated FDR," since FDR is a statistical expectation. Maybe just say "experimentally validated proportion of discoveries that are accepted but deemed to be incorrect."

Otherwise, you have done a good job of responding to my various suggestions and questions.

(Remarks on code availability)

Reviewer #3

(Remarks to the Author)

All issues raised during review have been thoughtfully addressed.

(Remarks on code availability)

We appreciate the time and effort that the reviewers have dedicated to our manuscript. We are grateful for their comments and suggestions and we have been able to incorporate changes to reflect all of the suggestions provided by the reviewers. The main changes are:

- Comparison of xiSEARCH, xiSEARCH+Prosit-XL, and XlinkX on the cytomegalovirus dataset.
- Evaluation of Prosit-XL and pDeepXL performance on the synthetic peptide dataset.
- Comparison of minimum, mean, and maximum approaches for selecting the final CSM score, demonstrating the efficiency of the minimum approach in distinguishing true-positive TTs.
- Estimation of expected true positive matches using the formula: $\#TT - (\#TD - \#DD)$ after rescoring across all datasets.
- Rescoring for all analyzed datasets in this study using the final version of Percolator.

We have highlighted the changes within the manuscript. Here is a point-by-point response (our response highlighted in green) to the reviewer's comments and concerns.

Review 1 (Remarks to the Author):

Kalhor et al. present Prosit-XL, an extension of the spectral prediction framework Prosit to cross-linked peptides. Prosit-XL is designed to support cleavable and non-cleavable reagents alike. To compensate for the somewhat limited datasets for training that are available (compared to linear peptides), a transfer learning strategy was implemented. Impressive results as judged by spectral angles and Pearson correlation were achieved for all data types. Prosit-XL was integrated into a rescoring approach using the Oktoberfest and Percolator pipelines and first applied to existing benchmark data sets, where it showed promising performance. Subsequently, spectral prediction and PSM rescoring was applied to more complex data sets to demonstrate that it can recover increased numbers of protein-protein interactions and higher cross-link coverage between interaction partners alike.

Taken together, Prosit-XL is a very promising enhancement to Prosit and might be the best performing approach for the prediction of MS/MS spectra in the cross-linking field reported so far. The selected applications presented in this manuscript already show the potential of spectral prediction and rescoring for cross-linking applications, where it should be even more beneficial than for unmodified, single-chain peptides because of poor fragmentation and/or poor spectral quality due to the low abundance of cross-linking products.

We appreciate the positive feedback.

General comments:

Throughout the manuscript, the authors use results from several cross-link search engines, including pLink, XlinkX, Scout, and xiSearch. While this to some extent highlights the general applicability of their approach, it comes at a risk of convoluting the effects of search engine performance and rescoring performance, at least in some cases. FDR control strategies will also

be different between the pipelines, and both pLink and XlinkX have shown large deviations between predicted and actual FDRs in several studies. As an example, the CMV dataset compares XlinkX with xiSearch + Prosit-XL. Why not re-search the data with xiSearch first without applying Prosit-XL?

We understand the concern regarding the potential conflation of search engine performance and rescoring performance. To address this, we have now included the results of xiSEARCH + xiFDR without Prosit-XL for all datasets, specifically for the CMV dataset (Supplementary Fig. S5b). This allows for a direct comparison of xiSEARCH + xiFDR, xiSEARCH + Prosit-XL + xiFDR, and XlinkX, ensuring a clearer evaluation of the individual contributions of each method. For the CMV dataset, xiSEARCH + Prosit-XL + xiFDR identifies the highest number of CSMs and PPIs, followed by XlinkX, and then xiSEARCH + xiFDR.

To clarify, while we initially hoped to be able to rescore other search engine results, this is not always possible e.g. due to the lack of decoys in the search results or lack of access to unfiltered search results. However, to show an improvement over the published state of the art, we attempted to always compare against the original results highlighted in the original papers.

In several examples, the numbers go up with respect to CSMs, PPIs etc., although the actual FDR appears to increase as well (for cases where ground truths or entrapment designs are available), see data reported in Figs. 3a and 4a. I feel it would make more sense to try to adjust the thresholds in a way to make the actual FDR comparable between the +/- Prosit-XL workflows, which should be feasible?

We found this a fair point for both the synthetic peptide dataset and the large-scale *E. coli-M. pneumoniae* dataset, where the actual FDR for the synthetic peptide dataset (Fig. 3b) and the lower bound estimate of the actual FDR for the large-scale dataset (Fig. 4b) slightly increased after rescoring. In response, we conducted a new analysis by applying an actual FDR of 1% to both the synthetic peptide dataset and the combined *E. coli-M. pneumoniae* dataset. The corresponding Figures have been included and referenced in the main manuscript. Briefly, for the synthetic peptide dataset, rescoring increased the number of identified CSMs from 1,175 to 1,216 and for peptide pairs from 641 to 651 (Supplementary Fig. S3b). For the large-scale dataset, we observed an even bigger improvement after rescoring, with the number of identified CSMs increasing from 1,711 to 3,389, for peptide pairs from 1,147 to 2,378, and for PPIs from 390 to 678 (Supplementary Fig. S4a).

In the introduction, the authors mention pDeepXL as previous work on the topic. Is it possible to somehow compare the performance of pDeepXL with Prosit-XL? If not, please explain why not.

In response to this point, we have added a new panel (d) to Supplementary Figure S2, comparing the prediction performance of pDeepXL and Prosit-XL. Additionally, an explanation has been added to the Methods section to clarify the comparisons. Briefly, we evaluated the performance of both models on synthetic peptide datasets as an external, unseen dataset. Prosit-XL demonstrated higher accuracy, achieving an SA of 0.82 and a PCC of 0.95, compared to pDeepXL, which achieved an SA of 0.74 and a PCC of 0.85, on a dataset that used DSSO as a

cleavable crosslinker. However, both models showed similar performance on a synthetic dataset that used DSS as a non-cleavable crosslinker, where Prosit-XL achieved an SA of 0.75 and a PCC of 0.90, while pDeepXL achieved an SA of 0.76 and a PCC of 0.87.

Although in the end it probably does not matter so much, how is the order of peptides A and B defined? Based on the search engine output, or based on a definition of Prosit-XL? In line 162, the authors state that "on average, peptide B is shorter ...". If the longer peptide is always designated as A/alpha as is commonly done, then this should be an obvious conclusion (unless the two peptides have the same length).

Generally, the order of the peptides is defined by the search engine. For xiSEARCH, the order is determined by the number of fragments explained - with fragments associated to the peptide in which the actual fragmentation event occurred. While this often biases the assignment toward the longer peptide being designated as peptide A, it is not a fixed rule. Different search engines may employ different metrics for determining this order.

The main point we intended to convey in the relevant section is that, in general, shorter peptides are less challenging for Prosit-XL to predict than longer peptides, which has been observed for linear peptides as well. It is important to note that the order of peptides A and B does not impact Prosit-XL's performance, the features generated by Oktoberfest, and the final Percolar score in any way. To address this, we have provided further clarification on this in the Methods section of the manuscript.

The figures are very dense, and the smaller text is hardly legible even at 200% zoom factor. One particular example is Fig. 4a, where I could find a typo ("Final database") only after zooming in even further. Some of the SI figures are even worse in this regard.

We have increased font size for readability in both the main and Supplementary Figures.

Additional comments:

Line 41: A frequently given argument is that the n^2 problem / search space expansion "increases the chance of false positive identification". I would argue that as long as FDR is controlled properly, the increased search space will rather increase false negatives because more stringent score thresholds need to be applied. Should tools such as xiFDR not handle that appropriately?

We agree with the reviewer. What we intended to convey is that an increased search space typically increases the chances (and score) of false positives (prior to FDR filtering). This in turn requires a high score cutoff to maintain a fixed FDR resulting in higher false negatives (after FDR filtering) and lower number of true positives. We have updated the introduction of the main manuscript to avoid this confusion.

Line 190: From this discussion, it appears that features were introduced to Oktoberfest that are specific to cross-linked peptides. If so, they should be discussed somewhere in the manuscript.

We apologize for causing this misunderstanding. In fact, no additional or cross-linking specific features are generated by Oktoberfest. Oktoberfest generates, as it does for linear peptides, intensity-based features, here, separately for peptide A and peptide B. This is possible because Prosit-XL predicts spectra for each peptide independently. To clarify this and avoid the confusion, we modified the relevant section in the results. However, to avoid too many references and backtracking, we have added a new Supplementary Information (Supplementary Information 1) listing all features and their explanations.

Line 207: Although I agree that the strategy to use the minimum score of the two peptides for assessment of quality is sensible, I do not think that Supplementary Information 1 provides a real justification or adds more to the discussion.

The approach of using the minimum score of two peptides to assess the quality of a CSM has been explored in previous search engines. We have updated the introduction to include relevant citations discussing this method. Additionally, we have added the results of further plots in Supplementary Information 2, where we tested the mean and maximum score approaches. Briefly, the results suggest that using the minimum score of two peptides is a more efficient method for differentiating between true-positives TTs and false-positive TTs. While these results do not prove that the minimum score is the best approach, they show consistent evidence that the minimum score performs well across various scoring strategies.

Line 440: b1 (2+) is listed twice, probably it should say (3+) in one case.

This has been fixed.

Line 861: Although this is a paper about AI, the proteins are probably still viral rather than virtual ...

We thank the reviewer for spotting this :-)

SI1, line 6 of the paragraph: Why do target-target matches also comprise FP-FP matches?

We would like to clarify that a "target" does not necessarily equate to being correct. Targets represent the proteins that are expected to be present in the sample, but this does not guarantee that every match to a target protein is correct. As a result, each target can either be a true positive (TP) or a false positive (FP). Consequently, target-target matches can represent a variety of cases: TP-TP, TP-FP, FP-TP, or FP-FP matches.

Review 2 (Remarks to the Author):

This paper describes a pipeline for carrying out database search on mass spectra derived from cross-linked peptides. The key innovation is to integrate into the scoring various features produced by a deep learning model that predicts fragment ion intensities, which yields a large boost in statistical power. Another claimed innovation is a way to do FDR control using Percolator in this setting. I like several features of this work. For example, the approach to use linear peptides to estimate effective collision energy for cross-linked peptides and then use the estimated CEs for calibration (p. 3) is sensible and seems to work well. Furthermore, the model design, and particularly how previous models were adapted for transfer learning (p. 4 and Figure 2a), is very elegant. Overall, the work appears to be significant, but I also have some significant concerns that are outlined below.

We are grateful for the positive feedback.

One concern is with respect to novelty. The manuscript mentions that there are several existing methods that use post-processors for MS-XL analysis (line 52), but there is no discussion of the relative merits of their approaches relative to the one outlined here. Indeed, I think the idea of scoring two PSMs separately and then taking the min, which is presented as a key idea here, is already part of Kojak. More generally, what seems to be missing is a description of related methods, pointing out their differences, especially with respect to Prosit-XL.

We absolutely agree that using the minimum score has been investigated in previous studies, such as Kojak. We have clarified this in the introduction. However, our understanding is that the authors of the Kojak paper used the minimum score as one of the features for Percolator, which appears to be run at CSM level. The novelty of our approach is that we run Percolator at the PSM level, rather than the CSM level, and finally take the minimum (Percolator) score as the final CSM score which is subsequently used for FDR estimation by xiFDR. This was possible because Prosit-XL generates predictions for each peptide separately. We believe this is substantially different to the way it is described in Kojak as Percolator is tasked to optimize the aggregation of features into a single score that reflects the confidence of the match in our approach.

The second concern is with respect to the validity of the approach. One of the main claims here seems to be that the proposed pipeline carries out "accurate and precise FDR estimation." I have several critiques related to this claim.

First, in the context of linear peptide analysis, it has already been established that PSM-level FDR control is problematic, because the decoy peptides do not adequately account for "neighbor" peptide relationships in the target database, where those neighbors arise due to homology (<https://arxiv.org/abs/1501.00537>). As such, I think that all of the analysis here should be done at the (cross-linked) peptide level.

We generally agree with this point. However, in our pipeline, FDR estimation is handled by xiFDR, which by default uses 'unique CSMs' for FDR estimation rather than all CSMs <https://github.com/Rappsilber-Laboratory/xiFDR/blob/master/README.md>.

To avoid misunderstandings, we clarified this further and have added a sentence in the section where we explain our rescoring pipeline: 'Finally, the CSMs and their corresponding scores are submitted to xiFDR for FDR estimation (Methods).' Further details are provided in the Methods section.

Second, and on a related note, I could not find any indication of what version of Percolator was used for this analysis. This is important not only for reproducibility, but because the recent version fixed a longstanding problem with Percolator's FDR control. An important feature of the current version of Percolator is that it does FDR control at the peptide, rather than PSM level.

We are sorry for this oversight. We used Percolator version 3.6.1, which has been added to the Methods. Further, since this version does not contain the most recent changes, we performed rescoring with version 3.7.1. and compared the results to version 3.6.1 (new Supplementary Figure S6). Briefly, we did not observe any substantial changes indicating that this problem did not affect cross-linking. This is likely due to us using xiFDR for FDR estimation rather than Percolator, which already avoids leakage on peptide-level as best as possible.

On this point, we want to clarify that Percolator is only used for PSM score assignment and not for FDR estimation or control. FDR estimation is performed by xiFDR, which is applied conservatively across all levels, including on peptide pairs and PPIs. To avoid misunderstandings, we clarified this further and have added a sentence in the section where we explain our rescoring pipeline: 'Finally, the CSMs and their corresponding scores are submitted to xiFDR for FDR estimation (Methods).' Further details are provided in the Methods section.

Third, and perhaps most worryingly, the claim in the abstract that the proposed method does a good job of FDR control is belied by the results of your first experiment. On p. 6 we learn that, at a nominal FDR threshold of 1%, the proposed pipeline yields an error rate of 1.18% at the CSM level and 1.65% at the peptide pairs level. It is true that this is much better than the reported results for competing methods (reported later in the paragraph to be 2.2-5.7%). But it still does not constitute valid FDR control.

We agree with the reviewer that our phrasing was too optimistic. We have adapted this. One possible explanation for the observed deviation in the synthetic peptide dataset is that there may be some issues with the dataset itself. However, we have no real evidence to support this. Additionally, the small dataset size (only 100 synthetic peptides) certainly presents challenges for accurate FDR estimation. We intentionally left the results of this experiment in the paper as we believe this is still an important dataset to evaluate our approach, despite the questions it raises. We have additionally dedicated a small section of the discussion on this point and the potential shortcoming of a machine-learning focus approach when dealing with small(er) datasets.

Fourth, more generally, I missed in the manuscript any theoretical argument for why the proposed method should actually control the FDR. We are only provided with empirical results on several datasets, which is obviously insufficient to guarantee FDR control in general. There is a pretty convincing proof of FDR control in the cross-linked setting (Walzthoeni et al.), albeit one that does

not incorporate a machine learning post-processor. I am surprised that none of this theory is applied in the current setting.

For FDR estimation/calculation, we rely on xiFDR, a well-known tool in the field. xiFDR estimates FDR using the formula derived in the mentioned reference (Walzthoeni et al.) and thus operates within the bounds of the theoretical argument.

Providing a theoretical proof that the proposed approach, particularly incorporating machine learning as post-processors, may not be possible. To address the concerns, we we have added a new Supplementary Figure S4b that shows the number of expected true positive matches by calculating $\#TT - (\#TD - \#DD)$. While not being a proof for correctness, the Figure clearly shows that the number of estimated true positives is highest in the high scoring region and, as expected for Percolator, drops to near 0 around a score of 0. At lower score, the number of estimated true positives stays around 0, an indication that the rescoring approach and the use of machine learning did not incorrectly introduce a bias (i.e. separating targets from decoys) and that the decoys (TD and DD) provide a good estimate for the number of false positive target-targets.

For transparency, we want to highlight here that the target/decoy distribution for Scout (Supplementary Figure S3a, synthetic protein dataset) still confuses us. In contrast to all other datasets, we observe a substantially higher number of decoys identified in the low scoring region, compared to the number of expected false positive targets. We have reached out to the developers of Scout to understand Scout's assumptions and implementation, but did not get a response yet. While the height of the decoy distribution is unexpected, the mode and variance is very much in line with the score distribution of the expected false positive targets. We have spent a considerable amount of time looking into this, however, we have to concede and hope the reviewer agrees that fixing this (in Scout) is beyond the scope of this manuscript.

In general, I think it's unfortunate that you've chosen to adopt the term "actual FDR" (line 212). In the statistics literature, I think the false discovery proportion (FDP) is the accepted term here, where the FDR is the expectation of the FDP.

We generally agree with the reviewer but decided to keep it as "actual FDR" and expanded our definition in the caption of the Figure. Unfortunately, even in "ground truth" data, we do not actually know if a spectrum match is in fact a true positive. All we can assess is if a spectrum is assigned to a cross-linked peptide that we expect to see, rather than a version we do not (to the best of our knowledge) expect to observe. The proportion of false positives among our expected true positives is unknown. Likewise, as indicated in our concerns about the synthetic peptide dataset, we may also incorrectly label a false positive as such, because we have to assume that the experiment was done without any errors (e.g. mis-labeling or incorrect pipetting schema).

You also switched from "actual FDR" to "real FDR" on line 249.

We use consistent nomenclature now.

I was struck by the fact that, in the section on large-scale datasets, you talk about using a method that is "similar to entrapment searches." Did you consider using one of the established entrapment

protocols. For example, there is a recent preprint (Wen et al. 2024) that outlines several such methods. Why not use one of those here?

We did not consider using one of the established approaches as the employed double entrapment approach has the distinct advantage that it catches more cases of decoy to false positive imbalances that can not be seen with traditional entrapment approaches. This has been described in Fischer et al. 2024 (<https://doi.org/10.1038/s44320-024-00057-2>). Of note, the estimate is a lower bound as estimating an upper bound is more complex.

Minor points:

line 52: Kojak uses Percolator, not PeptideProphet, as claimed here.

Thank you for catching this mistake. We have modified the text.

line 64: "indicating" should be something like "yielding"

Done.

line 71: You say that Prosit-XL "retained" its CE-awareness. I don't understand how it can "retain" anything, since Prosit-XL was introduced in this paper.

We intended to say "retained its CE-awareness from Prosit", but omitted the "from Prosit". To clarify, we used the word "inherited" now.

line 121: I think you should start a new paragraph here.

Done.

line 158: I found this long list of ten SA values awkward to parse. I would recommend just saying something like "improved by XX--XX%", since the specific values are reported in the accompanying figure.

Modified as recommended.

line 170: "remarkable" -> "remarkably"

Fixed.

line 232: What is "FP"?

Changed to "Filtered at 1% FDR, both pipelines produced less than 1% false positives (FP)".

Review 3 (Remarks to the Author):

Kalhor et al. present a study that uses transfer learning to apply Prosit to the problem of intensity prediction for CSMs – crosslinked spectrum matches. These CSMs are collected from crosslinked peptides arising from a range of different crosslinking scenarios (synthetic crosslinked peptides to in situ crosslinking), and using different styles of crosslinker (cleavable and noncleavable). The success of intensity prediction is parlayed into an elegant rescoring approach that leads to demonstrable improvements in performance across a range of XL-MS problems.

The paper is excellent, and the authors are commended for making the underlying methodology clear and easy to understand. It is well written. Prosit-XL provides a considerable performance boost that the community will welcome.

We are very grateful for the positive feedback.

I have only two noteworthy comments to make, and then a few very minor ones:

General comments:

1. Some additional commentary should be provided around the surprisingly high CSM losses shown in figure 3b,c (and to a lesser degree in figure 4b). This isn't the same as the imperfect overlap in figure 5b, where it is expected that XlinkX and xiSEARCH won't generate the same results. But in figure 3 and 4, the comparison involves xiSEARCH with/without PrositXL. Are the losses because of misidentifications in the search that doesn't use PrositXL? That doesn't make sense though, given the highly controlled FDR. Basically, why are so many high quality CSMs dropped?

An important point to highlight is that xiFDR outputs only unique CSMs rather than all identified CSMs. Specifically, for each unique combination of peptide pair sequence, modifications, link sites, and charge state, xiFDR retains only the top-scoring CSM. This inherent filtering contributes to the observed losses.

To investigate this, we focused on the synthetic peptide dataset as a case study. As shown in Figure 3b, we observed a loss of 333 CSMs. To better understand this, we ran xiFDR twice — once with Percolator output and once with xiSEARCH output — both at an FDR of 100% (df_percolator_100%, df_xisearch_100%) and analyzed the results. As expected, xiFDR removed some CSMs even at FDR = 100% in both cases. Interestingly, of the 333 lost CSMs, only 24 appeared in the df_percolator_100% output, while the remaining 309 were entirely absent. This explains why they were not recovered at an FDR of 1%.

This suggests that the losses are not due to misidentifications but rather stem from how xiFDR selects and retains top-scoring CSMs while filtering out others, even before applying an FDR threshold.

2. The authors' use of the term PPI is misleading in several locations. This comes up particularly in the treatment of the HCMV datasets, but I think other places as well (hard to tell). They define inter-protein crosslinks as “between-links” and intra-protein crosslinks as “self-links” but then declare PPIs to equal self + between links (line 277). This is not correct. Please use PPIs to

represent only the inter-protein data and the clearly unambiguous intra-protein data (i.e., only where the linked peptides share sequence). All others should be assumed intra-protein linkages and thus not a PPI, just a protein. It is important to be consistent with other papers on this point.

We are sorry for this oversight and agree with the reviewer. We have updated the manuscript and related Figures accordingly. To avoid any further misunderstanding, we have classified the observed peptide pairs into inter-protein interaction, intra-protein interaction (where the linked peptides are a shared sequence), and intra-protein connection.

Very minor matters:

1. Line 268ff: it is more correct to say you are maintaining control over the FDR, not the FDR calculation.
2. Line 345ff: underpins the value OF orthogonal information...
3. Line 496: spelling of xiSEARCH
4. Line 526: spelling of replicate

All issues have been addressed.